# NETWORK OF THESEUS (LIKE THE SHIP)

## ABSTRACT

A standard assumption in deep learning is that the inductive bias introduced by a neural network architecture must persist from training through inference. The architecture you train with is the architecture you deploy. This assumption constrains the community from selecting architectures that may have desirable efficiency or design properties due to difficulties with optimization. We challenge this assumption with Network of Theseus (NoT), a method for progressively converting a trained, or even untrained, guide network architecture part-by-part into an entirely different target network architecture while preserving the performance of the guide network. At each stage, components in the guide network architecture are incrementally replaced with target architecture modules and aligned via representational similarity metrics. This procedure largely preserves the functionality of the guide network even under substantial architectural changes—for example, converting a convolutional network into a multilayer perceptron, or GPT-2 into a recurrent neural network. By decoupling optimization from deployment, NoT expands the space of viable inference-time architectures, opening opportunities for better accuracy–efficiency tradeoffs and enabling more directed exploration of the architectural design space.

## 1 INTRODUCTION

In machine learning research, we tend to assume that training is coupled with inference: If a specific architecture is trained to solve a task with specific inductive biases and computational mechanisms, that same architecture should be used at test time. This assumption is embedded in common practice: neural architecture search (NAS) discovers an efficient inference-time architecture that is then trained and deployed (Zoph & Le, 2016; Real et al., 2019; Liu et al., 2018). Compression pipelines prune and quantize a trained model to meet deployment budgets while preserving the trained structure (Han et al., 2015; Jacob et al., 2018). This assumption applies even in settings like distillation, where teacher networks train student networks designed explicitly for inference (Hinton, 2015; Gou et al., 2021; Romero et al., 2014; Huang et al., 2022). A few lines of work partially relax this coupling by training a supernet and selecting subnets for deployment (Cai et al., 2019; Yu et al., 2018), but the resulting inference models remain architectural subgraphs or variants of the original training network rather than cross-architectural conversions.

We challenge this premise that the architecture used for training must be the architecture used for inference. Decoupling the architecture used for training from that used for inference would enable models to be trained with large, optimization friendly architectures and converted into lighter architectures for efficient inference on edge devices. This would also enable controlled exploration of inductive biases by comparing architectures without confounding optimization difficulty.

Large-scale analyses demonstrate that very different architectures (e.g. CNNs vs Transformers) converge toward similar internal representations as they scale (Huh et al., 2024; Han et al., 2023; Li et al., 2015; Conwell et al., 2024). Furthermore, classic observations of representational alignment across independently trained networks (Raghu et al., 2017; Kornblith et al., 2019) suggest that sufficiently expressive, distinct architectures can implement the same input–output functions. This statement of universality is also supported by theoretical work that shows that functions that are Turing-computable can be approximated by any neural network (Poggio & Fraser, 2024). However, the real challenge is not whether such functions exist, but how to reach them through optimization. This view implies that architectural priors are not constraints that must persist at inference, but are primarily training biases – scaffolds designed to guide and stabilize optimization.

Motivated by these perspectives and potential benefits, we introduce *Network of Theseus* (NoT), a part-by-part conversion procedure that starts from a given guide architecture and progressively replaces its components (e.g. layers) with target modules. Our name for this procedure is a reference to Plutarch's Ship of Theseus paradox (Plutarch, 100–125), which asks: once all the decaying planks of a ship are replaced (and none of the original remain), is it still the same ship?

At each replacement stage, we instantiate a target module and optimize it to match the activations of the guide network using a representational distance function (Kornblith et al., 2019; Cristianini et al., 2001; Cortes et al., 2012). This optimization transfers priors between networks at a lower level, e.g. at the layer level rather than at the architectural level (Subramaniam et al., 2025). After the conversion is complete, the resultant target architecture is trained end-to-end on the downstream task. This staged alignment decouples training from deployment: we can train with one architecture and convert it into an entirely different one for test-time use, aiming to preserve the original performance as closely as possible.

Using NoT, we show broad architectural conversions: convolutional (CNN) models to fully connected MLPs with low-rank linear layers, vision transformers with multihead attention to token-wise MLPs, and transformer language models to Elman RNNs. Interestingly, we observe even untrained guide networks contain useful inductive biases; NoT transfers these effectively, yielding performance comparable to using trained guides. Furthermore, we find that NoT can convert from larger to smaller networks and that converting from a larger to a smaller architecture (e.g., a deeper ResNet to a shallower one) can at minimum preserve performance. We show our conversions using centered kernel alignment (CKA) and further validate our findings by introducing a new kernel-based similarity metric, differentiable mutual nearest neighbors (D-MNN). Designed from the metric introduced in Huh et al. (2024), D-MNN exploits local geometric structure for comparing representations.

NoT has both practical and theoretical implications for architectural transfer and design. With trained or untrained guides, we can convert to architectures that are more efficient or better aligned with deployment constraints or theoretical desiderata. Because NoT aims to preserve performance during conversion, it enables discovery of inference architectures with accuracy–efficiency trade-offs not previously attained, including transfers from larger (even untrained) guides to smaller targets. NoT implies that designing and training an architecture with simple optimization dynamics is not necessary: we can now design and deploy neural networks where constraints matter, expanding architectural choice at inference time without resolving the full supervised learning problem from scratch for every candidate design.

Our contributions are as follows:

1. We introduce the Network of Theseus (NoT), a method to part-by-part convert from one architecture to another using representational alignment.
2. We demonstrate NoT across a wide variety of architectural conversions such as converting convolutions in ResNet-18 to linear layers to create an MLP or converting attention in GPT-2 to RNN layers to create a Deep RNN. We find that across all conversions, we preserve performance of the original architecture despite dramatic architectural shifts.
3. Surprisingly, we find that we can perform the same conversion starting from an untrained network architecture, demonstrating that we can transfer inductive biases to our target networks.
4. We validate our findings using representational similarity metrics like CKA and introduce a new metric based on a differentiable version of mutual nearest neighbors, D-MNN, to further validate our findings as general and invariant to metric.

## 2 RELATED WORK

**Architectural Transfer via Linearizing Transformers**: NoT is a general method that builds on work applying cross-architectural distillation for linearizing transformers (Mercat et al., 2024; Zhang et al., 2024a;b; Bick et al., 2024). These works focus on specifically distilling multihead attention to either SSMs (Wang et al., 2024; Bick et al., 2025) or linear attention. Most approaches involve training linear attention layers or SSMs to approximate softmax attention via MSE. Some show that linearizing can specifically scale to very large transformers (Zhang et al., 2024a) e.g. 7B - 72B LLMs without considerable effect on performance.

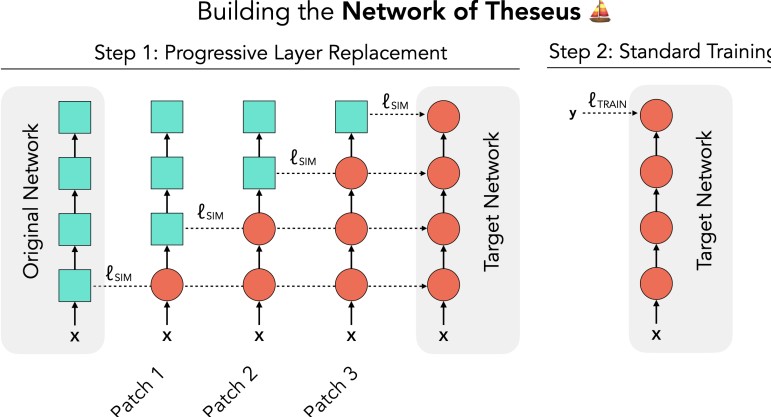

Figure 1: **Network of Theseus**. A network can be converted to any desired target network by replacing each piece of the original network incrementally, part-by-part. Each original part is replaced by optimizing the representational alignment, $\ell_{SIM}$, of the target part to the original part. After all original parts are replaced, only the target network remains and can be trained on any downstream task (i.e. standard training).

These papers are significantly different from NoT, both in scope and approach. We greatly expand architectural transfer with linear attention or SSMs to transfers that are not dependent on having the same computations in order to use MSE. Our representational distance functions allow for any architectural pairing as well, making our conversions much richer. While previous work linearized with only linear attention, we can now focus on other layer replacements such as converting convolutional layers to linear layers, etc. NoT also makes methodological changes to how layer swaps are applied, considering changes that are staged and progressive rather than all-at-once. Our comparisons show that this progressive replacement is important, especially when architectures are different.

**Model Distillation and Compression**: Distillation (Hinton, 2015; Gou et al., 2021; Sanh, 2019; Hsieh et al., 2023; Tian et al., 2019; Chen et al., 2021; Lin et al., 2020) transfers knowledge from a teacher model to a student model by introducing a new component to the loss function that forces the student model to behave like the teacher model (Kim et al., 2021; Zhou et al., 2021). This usually consists of penalizing the KL-divergence between the logit predictions of the student and teacher model. Methods have been proposed that use CKA as an alignment approach between representations of two networks or with representations in the brain with notable improvement in network performance (Saha et al., 2022; Dapello et al., 2022).

Other works also transfer knowledge across models while compressing information. For example, BERT-of-Theseus (Xu et al., 2020) uses the "Ship-of-Theseus" framing to compress BERT into a smaller architecture using stochastic replacement schedule during supervised training. NoT significantly expands on BERT-of-Theseus to include multiple replacement schedules, a wider range of architectures, and include untrained starting architectures. Similarly, other papers gradually convert layers to identity functions in order to shrink the given architecture (Chen et al., 2023), where adjacent convolutions are merged for inference. This uses a within-architecture training trick to prune non-linearities and fuse layers.

We distinguish NoT from distillation and compression. NoT aligns computations and can transfer inductive biases from one architecture to another. Like prior work (Subramaniam et al., 2025; Ulyanov et al., 2018; Zhong & Andreas, 2024), this allows us to use completely untrained guide networks as part of our conversion, although our replacement strategy in NoT is distinct from this previous work. Doing the same with distillation would lead to much worse results (Subramaniam et al., 2025). Furthermore, to our knowledge, the stage-wise replacement of NoT has not been proposed in previous work.

**Representational Convergence**: Our paper builds on prior investigations into representational alignment across different network architectures. Prior work has seen both behavioral and representational convergence across diverse neural network architectures. For example, prior work has found that networks trained with self-supervised and supervised have similar errors, indicating behavioral alignment (Geirhos et al., 2020). Many other prior works have discovered similar representational

alignment through different similarity metrics such as kernel alignment (Huh et al., 2024; Han et al., 2023) and model stitching (Bansal et al., 2021). These prior works have discovered representational convergence across different architectures, learning tasks, and even modalities. This representational convergence across architectures and learning tasks have been exploited in prior work to improve network performance. For example, representational alignment is used to fix architectures that are traditionally ill-suited for certain tasks due to optimization failures or diffusion models with inconsistent representations (Yu et al., 2024). Similar to these, we use representational convergence in NoT to convert across architectures progressively.

## 3 METHODS: NETWORK OF THESEUS (NOT)

**Network of Theseus (NoT)** provides a general procedure for replacing parts of a source ("guide") network with parts of a target architecture while preserving internal representations. At any time, a subset of components is replaced; we then train the replaced subset to *match* the guide's activations, stage by stage, according to a **replacement schedule**. Figure 1 illustrates our default schedule—**progressive replacement**, which we find works best empirically. Figure 2 compares alternative schedules. NoT is task-agnostic. After the replacement completes, we finetune end-to-end for the downstream objective (e.g., image classification or language modeling).

Let $f^G(\cdot)$ be a fixed guide network with $k$ layers and let $f^T(\cdot; \theta)$ denote the under-construction target network obtained by replacing some of the guide's layers with new modules (e.g., Conv2d→Linear, attention variants, or block-level substitutions). We assume that both networks share input/output interfaces so that they can be run on the same inputs $\boldsymbol{x} \in \mathbb{X}$. For a mini-batch from train, let $\boldsymbol{A}_i^G(\boldsymbol{x})$ and $\boldsymbol{A}_i^T(\boldsymbol{x}; \theta)$ be the activations extracted at layer $i$ from $f^G$ and $f^T$, respectively. In practice, for similarity computation, we flatten spatial/token dimensions so each row corresponds to a sample.

We write $g(\cdot, \cdot) \in [0, 1]$ for a representational similarity (e.g., linear CKA) and use its complement $\Delta(\boldsymbol{A}, \boldsymbol{B}) \triangleq 1 - g(\boldsymbol{A}, \boldsymbol{B}) \in [0, 1]$ as a **dissimilarity** to be minimized.

### 3.1 PART-BY-PART MATCHING OBJECTIVE

For illustrative purposes, we show how to apply NoT for a single layer $i$. The matching loss is

$$\mathcal{L}_i(\boldsymbol{\theta}_i) = \mathbb{E}_{\boldsymbol{x} \sim \mathcal{D}} \left[ \Delta\big(\boldsymbol{A}_i^T(\boldsymbol{x}; \boldsymbol{\theta}_i), \boldsymbol{A}_i^G(\boldsymbol{x})\big) \right], \tag{1}$$

where $\theta_i$ are the parameters introduced at location $i$ by the replacement. At a training *stage*, we optimize a set $I \subseteq \{1, \cdots, k\}$ of replaced layers jointly:

$$\mathcal{L}_I(\boldsymbol{\theta}_I) = \mathbb{E}_{\boldsymbol{x} \sim \mathcal{D}} \left[ \frac{1}{|I|} \sum_{i \in I} \Delta\big(\boldsymbol{A}_i^T(\boldsymbol{x}; \boldsymbol{\theta}_I), \boldsymbol{A}_i^G(\boldsymbol{x})\big) \right]. \tag{2}$$

Layers not in $I$ are frozen to prevent drift in the guide pathway.

**Replacement Modules**: NoT is agnostic to the specific replacement, requiring only shape compatibility. For Conv2d, we use an *equivalent low-rank linear* that applies to a flattened input and reshapes back to the expected output. Because of a large parameter count for the target linear layer, we first apply a linear layer to project the input to a lower dimensionality and then use another linear layer to project the desired output dimensionality, both of which are tunable. For attention, we support replacements with basic tokenwise MLPs, without any token-mixing, or RNNs. The framework also accommodates block-level replacements e.g., ResNet-50→ ResNet-18 basic blocks.

### 3.2 REPLACEMENT SCHEDULES

A **replacement schedule** is a sequence $\mathbb{S} = (I_1, \cdots, I_T)$ of index sets indicating which layers are trainable at each stage. NoT supports several schedules; we adopt **progressive replacement** by default due to its favorable stability–accuracy tradeoff.

**Progressive**: At stage $t$, we replace the $t$-th layer and set $I_t = \{1, \cdots t\}$ jointly optimizing all previously replaced layers together with the new one. This minimizes error accumulation while x feature distributions calibrated across the replaced prefix (Figure 1).

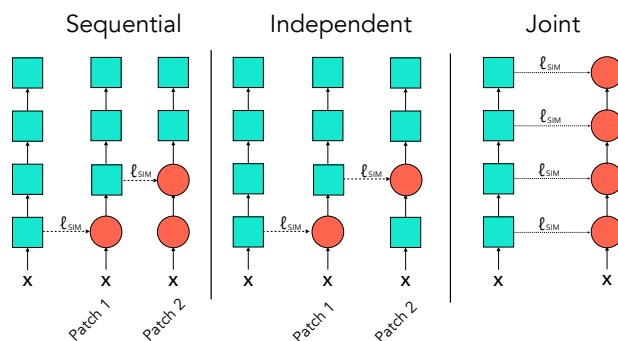

Figure 2: **Alternative replacement schedules**: *Sequential*: Each layer is replaced while holding each previously replaced layer frozen. *Independent*: Each layer is replaced independently, routing the input from the original layer below to the target layer above. *Joint*: Each target layer is trained jointly or simultaneously without any progressive conversion.

**Sequential**: Identical to progressive in the order of replacement, but we *freeze* all previously replaced layers and optimize only the newest one, i.e., $I_t = \{t\}$ with earlier replacements fixed. This can suffer from error accumulation because misalignment in earlier layers propagates forward (Figure 2).

**Independent**: Each layer $i$ is optimized *in place* using the guide's input distribution at that layer, without coupling to other replacements. After fitting all layers independently, we assemble the full target. This avoids sequential error accumulation but introduces distribution shift once the independently trained layers are composed (Figure 2).

**Joint**: Replace all $k$ layers at once and set $I_1 = \{1, \cdots, k\}$. This removes schedule bias but can be harder to optimize because later layers must adapt to rapidly changing early layers (Figure 2).

**Group-wise variants**: Beyond scalar layer indices, $I_t$ can denote *groups* (e.g., a ResNet group). Our approach supports progressive group introduction (e.g., ResNet-50 → ResNet-18) as well as progressive *alignment* where a smaller student unfreezes groups over time to match a fixed teacher.

### 3.3 SIMILARITY CHOICES AND TASKS

We instantiate $g$ with **linear CKA** or **differentiable mutual nearest neighbors (D-MNN)**, both of which are scale-invariant across rows (mini-batch samples). D-MNN is a representational similarity metric designed and introduced in this paper based on the nearest neighbor metric used in Huh et al. (2024). In this work, we were inspired by the locality property of the metric and aimed to make it differentiable in the hope that local structure in the representation space has structure that can be exploited and aligned. We refer to Appendix C.1 and Appendix C.2.

## 4 EXPERIMENTS

We apply NoT across a range of architectures and replacements to measure how well we can use our staged replacement across different replacement schedules. For all experiments, we compare using a trained and untrained guide network and separate training for representational similarity and task training into two stages.

**Tasks**: We consider image classification and language modeling in this paper. For testing image classification, we use the ImageNet dataset (Deng et al., 2009), measuring Top-1 performance on the pre-defined validation set. For language modeling, we use the Wikitext-103 dataset (Merity et al., 2016) where models must predict the next token given some context. We use a sequence length of 128 for all models and use the training, validation and testing sets defined by the dataset. We tokenize the text using the GPT-2 (Radford et al., 2019) tokenizer.

**Architectures and Replacements**: We study cross-family and within-family conversions spanning convolutions, attention, and recurrent computation. In all cases, layers are progressively replaced and optimized to match the guide's intermediate activations via representational similarity.

*ResNet-18→MLP*: We convert ResNet-18 (He et al., 2016) to a fully-connected MLP by replacing each convolutional layer with a linear layer. This removes the spatial priors of convolutional layers

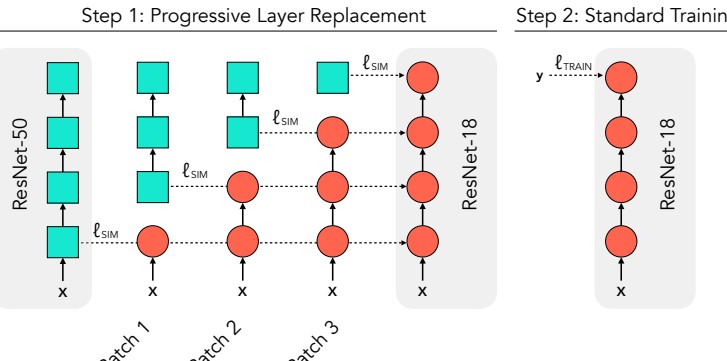

Figure 3: **NoT makes any-to-any architecture conversion possible.** Without the requirement of functional intermediate models, the target network can be any architecture. For instance, we convert the deeper ResNet-50 to the shallower ResNet-18. Blue squares are a set of multiple ResNet-50 blocks. These are replaced with a single ResNet-18 block. ResNet-18 blocks are added until we have a full ResNet-18 network.

such as receptive fields or translational equivariance. We use two low-rank linear layers (rank 256-1024) to keep the parameter size of the linear layers reasonable, as discussed in Section 3.1. Batch normalization recalibration details are in Appendix C.

*DINOv2→Patch-MLP*: We convert DINOv2 (Oquab et al., 2023) to a patch-wise MLP by replacing each multihead attention with a token-wise MLP that applies feedforward layers independently to each visual token. This eliminates cross-token communication, increasing efficiency at the cost of token interaction.

*GPT-2→RNN*: We replace GPT-2 (Radford et al., 2019; Vaswani, 2017) multihead attention layers with two Elman RNN layers per attention block. This tests whether sequential memory-based processing can substitute for parallel attention mechanisms.

*ResNet-50→ResNet-18*: We study architectural transfer rather than direct replacements that require matching input and output shapes. We map every four ResNet-50 blocks to one ResNet-18 block (blocks follow (He et al., 2016)), see Figure 3. We rebuild ResNet-18 block by block, independent of the guide architecture. At each stage the new block learns to match the representations of its four source blocks, testing whether NoT transfers depth based expressivity to a shallower model. This setting shows NoT can operate across any architectural conversion. See Appendix E.2 for details.

*GPT-2 Large→GPT-2 Small*: We conduct the same architectural transfer for language models. We map every three GPT-2 Large blocks to one GPT-2 Small block (blocks follow Radford et al. (2019); Vaswani (2017) and correspond to transformer decoder layers). We rebuild GPT-2 block by block, independent of the guide architecture. At each stage, the new block learns to match the representations of its three source blocks.

**Training**: For each setting, we train NoT with a guide network and the target replacements. We include a baseline with *naive replacement* where all guide network layers are replaced with the target modules from scratch for a fair comparison to understand what representational similarity provides to performance. During task training, all networks are trained with cross-entropy loss, without loss of generality. For all task training and representational similarity optimization, we use the AdamW (Loshchilov, 2017) optimizer. When optimizing representational similarity, we also incorporate gradient clipping due to unstable training. We use CKA for all settings and include D-MNN for the ResNet→MLP setting.

For all training, we use a consistent batch size of 256. Representational similarity metrics are affected by the batch size, specifically more samples allows the metric to better approximate similarity. Furthermore, we employ different learning rates across the progressive stages of representational similarity. We tune the learning rates carefully per layer replacement leading to different learning rates at every stage. Similarly, we use a different number of training epochs during representational similarity optimization, varying from 15 epochs to 100 epochs at every stage. For image classifica-

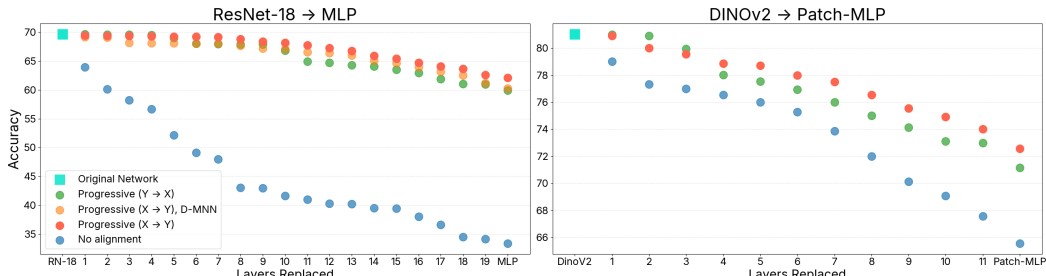

Figure 4: **Progressive layer replacement preserves performance across replacements.** We visualize progressive layer replacement across all patches. We apply a patch, reduce the CKA and finetune the resultant hybrid network until full replacement. This is compared with naive replacement with no CKA alignment. We compare forward replacement (X→Y, reverse replacement (Y→X) in both settings, and compare using D-MNN in the ResNet-18→MLP setting. Across all progressive replacements, we far exceed naive replacement with no alignment. We are not sensitive to replacement order.

| Original (Guide) → Target | Similarity Metric | Guide | NoT | Baseline |
|---|---|---|---|---|
| | | **ImageNet Top-1 Accuracy** (↑) | | |
| ResNet-18 → MLP | CKA | 69.66 | 62.12±0.42 (↑28.76) | 33.36±0.31 |
| ResNet-18 → MLP | D-MNN | 69.66 | 60.28±0.39 (↑26.90) | 33.36±0.31 |
| DINOv2 → Patch-MLP | CKA | 81.03 | 72.56±0.20 (↑7.01) | 65.55±0.45 |
| DINOv2 → Patch-MLP | D-MNN | 81.03 | 70.93±0.62 (↑5.38) | 65.55±0.45 |
| ResNet-50 → ResNet-18 | CKA | 76.13 | 73.81±0.26 (↑4.91) | 68.90±0.33 |
| ResNet-50 → ResNet-18 | D-MNN | 76.13 | 73.02 ±0.84 (↑4.12) | 68.90±0.33 |
| | | **Perplexity** (↓) | | |
| GPT-2 → RNN | CKA | 37.50 | 50.58±0.89 (↓70.61) | 121.19±3.12 |
| GPT-2 → RNN | D-MNN | 37.50 | 47.44±1.91 (↓73.75) | 121.19±3.12 |
| GPT-2 Large → GPT-2 Small | UCKA | 22.05 | 24.98±1.52 (↓3.03) | 28.01±1.02 |
| GPT-2 Large → GPT-2 Small | D-MNN | 22.05 | 26.02±1.22 (↓1.99) | 28.01±1.02 |

Table 1: **Network of Theseus vastly improves over naive replacement and preserves performance**. We compare NoT with standard training (baseline) and the original guide network performance. We find that NoT vastly outperforms naive replacement, close to 30% on ImageNet and 71 points on Wikitext-103.

tion tasks, we also use a warmup and cosine learning rate scheduler (Loshchilov & Hutter, 2016) with 5 warmup epochs. We apply task training for 100 epochs across all settings. Similarly, during task training, we sweep the learning rate and use a cosine and warmup scheduler with 5-10 epochs of warmup. More details on settings for learning rates and epochs of training are given in Appendix E. We did careful tuning and sweeps to ensure learning rate optimality.

After choosing the optimal learning rate, we then train all networks and settings with 5 random seeds to compute error bars. Our error bars are associated with the standard error across all seeds. We choose the seed-based average test perplexity associated with the epoch with the lowest seed-based average validation loss for the Wikitext-103 dataset.

## 5 RESULTS

**NoT significantly outperforms naive replacement**: We apply NoT to all of our previously described networks and summarize the results in Table 1. We find that across all settings, NoT improves performance over naive replacement by up to 30%. For example, we find that fully-connected MLPs can only be trained to achieve 33% accuracy. With NoT, we improve by 30% and identify a fully-connected MLP that is competitive with ResNet-18. Similarly, we are able to replace attention in vision transformers with Patch-MLPs while preserving accuracy, meaning that token communication is not necessary for downstream image classification, achieving competitive results on DINOv2. This holds for Elman RNNs, which become effective replacements for attention computations. Most excitingly, we find that NoT can be applied across similar architectures like ResNet-18. These results far exceed naive replacement and standard training. We also find consistent results with D-MNN as with CKA.

| Original → Target | Similarity Metric | Progressive | Joint | Independent | Sequential |
|---|---|---|---|---|---|
| | | **ImageNet Top-1 Accuracy** (↑) | | | |
| ResNet-18 → MLP | CKA | 62.12±0.42 | 57.98±0.28 | 45.10±0.23 | 30.73±0.14 |
| ResNet-18 → MLP | D-MNN | 60.28±0.39 | 57.66±0.22 | 45.81±0.06 | 31.90±0.33 |
| DINOv2 → Patch-MLP | CKA | 72.56±0.20 | 70.42±0.33 | 66.02± 0.47 | 47.15±0.32 |
| DINOv2 → Patch-MLP | D-MNN | 73.02±0.84 | 71.69±0.94 | 69.88±0.13 | 51.94±1.55 |
| ResNet-50 → ResNet-18 | CKA | 73.81±0.26 | 70.38±0.35 | 61.51±0.46 | 61.08±0.15 |
| ResNet-50 → ResNet-18 | D-MNN | 73.02 ±0.84 | 69.12±1.10 | 60.87±0.11 | 62.04±0.92 |
| | | **Perplexity** (↓) | | | |
| GPT-2 → RNN | CKA | 50.58±0.89 | 57.69±0.58 | 100.94±1.15 | 121.36±2.23 |
| GPT-2 → RNN | D-MNN | 47.44±1.91 | 56.66±1.55 | 115.96±1.65 | 136.98±0.54 |
| GPT-2 Large → GPT-2 | UCKA | 24.98±1.52 | 25.04±1.33 | 33.66±0.94 | 29.62±2.36 |
| GPT-2 Large → GPT-2 | D-MNN | 26.02±1.22 | 27.20±1.43 | 33.96±2.65 | 34.91±3.41 |

Table 2: **Progressive replacement outperforms other replacement schedules**: We compare progressive replacement with previously discussed replacement schedules, finding that our staged, progressive schedule is extremely useful. Joint replacement is the closest but likely requires significantly longer training.

| Original → Target | Similarity Metric | Guide | | Baseline |
|---|---|---|---|---|
| | | Trained | Untrained | |
| | | **ImageNet Top-1 Accuracy** (↑) | | |
| ResNet-18 → MLP | CKA | 62.12±0.42 | 60.85±0.48 | 33.36±0.31 |
| ResNet-18 → MLP | D-MNN | 60.28±0.39 | 60.90±0.69 | 33.36±0.31 |
| DINOv2 → Patch-MLP | CKA | 72.56±0.20 | 70.39±0.25 | 65.55±0.45 |
| DINOv2 → Patch-MLP | D-MNN | 73.02±0.84 | 68.15±0.96 | 65.55±0.45 |
| ResNet-50 → ResNet-18 | CKA | 73.81±0.26 | 71.61±0.42 | 68.90±0.33 |
| ResNet-50 → ResNet-18 | D-MNN | 73.02 ±0.84 | 70.22±1.18 | 68.90±0.33 |
| | | **Wikitext Perplexity** (↓) | | |
| GPT-2 → RNN | CKA | 50.58±0.89 | 58.26±0.43 | 121.19±3.12 |
| GPT-2 → RNN | D-MNN | 47.44±1.91 | 49.71±2.56 | 121.19±3.12 |
| GPT-2 Large → GPT-2 | UCKA | 24.98±1.52 | 25.10±1.97 | 28.01±1.02 |
| GPT-2 Large → GPT-2 | D-MNN | 26.02±1.22 | 26.15±2.43 | 28.01±1.02 |

Table 3: **NoT with untrained guide networks improves over naive replacement**. Can we apply NoT with untrained guide networks to transfer inductive bias? We find that untrained guide networks contain useful inductive biases and can improve over naive replacement with no alignment, given by baseline. NoT with untrained guide networks is competitive with NoT with trained guide networks.

Additionally, in Figure 4, we show training performance across our progressive layer replacements for ResNet-18 and DINOv2. We show the original performance, training with NoT and standard training with a naive replacement. When replacing with NoT, we consider progressive replacement in both the forward direction, from the first layer to the last layer, and the reverse direction, from the last layer to the first layer. This tests how sensitive results are to replacement direction. Across all layer replacements, we find that using representational similarity to incorporate a layer leads to stronger results in comparison to standard training. Reverse replacement leads to slightly worse performance, likely due to representational similarity drift. The improvement is significant across all layer replacements. We find that performance is lost on certain layers such as layer 13 in ResNet-18, showing that these layers are bottlenecks.

**NoT significantly outperforms training from scratch**: There have been significant efforts to train MLP models from scratch on ImageNet. Bachmann et al. (2023) achieved a performance of 51.5% on ImageNet with significant forms of training time data augmentation and test time inference augmentation. We achieve significantly high performance without any augmentation during training or test. We highlight this not to compare methodology but to highlight the difficulty of training an MLP from scratch compared to NoT, emphasizing the decoupling between the functions an MLP can represent as compared to the functions an MLP can acheive when training from scratch.

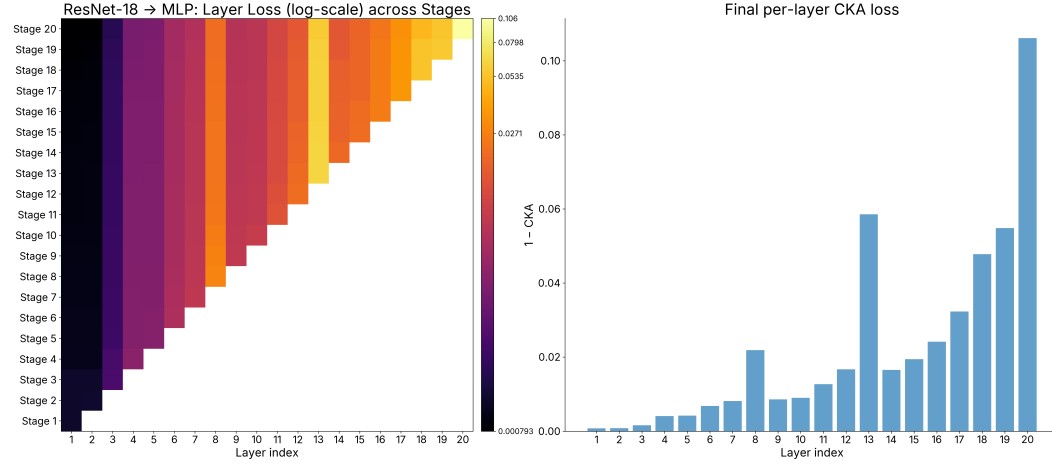

Figure 5: **Representational similarity across stages reveals difficult layers**: (left) We show CKA similarity losses (log-scaled) across all stages of progressive replacement. We see that CKA loss decreases across all stages for all layers. (right) We plot the final CKA loss for the last stage across all layers. We can identify bottleneck layers that are more difficult to align. Specifically, layers 6, 8, 13, and the final layers have higher loss. Layers 6, 8, and 13 are associated are associated with downsampling in ResNet-18.

**Alignment becomes more challenging at the end**: In Figure 5, we show our layer alignment loss and how it progresses across stages as well as the final loss for our ResNet-18→MLP experiment. We can identify similar bottleneck layers that are difficult to optimize similarity on such as layer 8 or layer 13. These layers are associated with downsampling the feature dimension in ResNet-18. Furthermore, we find that later layers are more difficult to align than earlier layers. This holds for our reverse progression from output to input and across all experimental settings with DINOv2 and GPT-2. We believe further work can be dedicated to make the later layers better aligned.

**CKA alignment predicts accuracy and improves with rank**: Does better representational similarity imply better task performance? We investigate the correlation using multiple checkpoints at our final progressive layer replacement, where we jointly tune all layers in the network to optimize CKA. This is shown in Figure 6. We find a direct relationship between average CKA and final task accuracy: higher average CKA correlates with higher final task accuracy. This indicates that most results can improve even further with NoT since a higher CKA alignment leads to stronger downstream performance. We also find that larger ranks tighten our CKA loss as well as improve our performance after task training. This also implies that a larger rank will lead to stronger results.

**NoT with progressive replacement significantly outperforms other replacement schedules**: In Table 2, we show comparisons of NoT between layer replacement schedules as discussed in Section 3.2 such as joint, independent or sequential replacement. We apply these different replacement schedules to all discussed architectural settings. We find that our staged progressive replacement significantly out-performs all replacement strategies. Joint replacement is competitive but ultimately cannot properly optimize layers and likely requires much longer training than allotted in comparison to progressive. As expected, independent replacement fails due to issues with distribution shift that are difficult to overcome at task training. Sequential replacement fails due to error accumulation.

**Untrained guide networks improve over naive replacements**: Surprisingly, we find that untrained guide networks are able to transfer useful architectural priors via NoT, leading to similar improvements. This is shown in Table 3. Across all settings, we see improvements with NoT even when the guide network is completely untrained. This shows that untrained networks have useful priors as noted in previous work (Subramaniam et al., 2025; Ulyanov et al., 2018; Zhong & Andreas, 2024). This distinguishes NoT from distillation given that distillation does not work with untrained networks. More importantly, we believe this has striking implications for architecture transfer. We highlight our result with ResNet-18 and ResNet-50. Converting from an untrained ResNet-50 to ResNet-18 results in a 3% performance increase. This suggests that depth and connectivity can act as transferable priors even without learned weights. We believe this has strong implications beyond NoT for distillation, where the assumption was that we would always need a trained architecture to train another trained architecture.

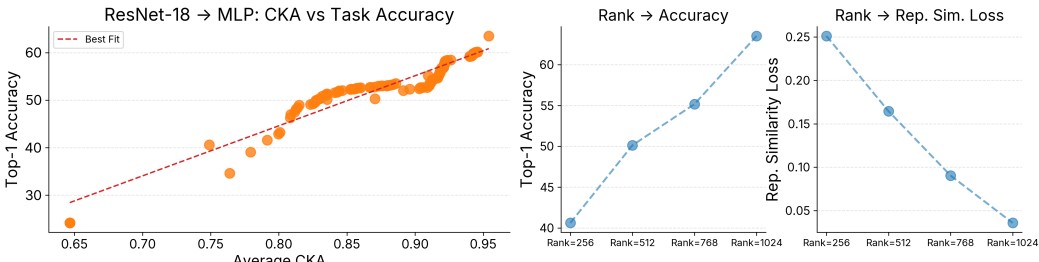

Figure 6: **NoT performance improves with larger CKA and larger ranks**: We compare final task performance of the ResNet-18→MLP setting over different final CKA alignment scores (left) and linear layer ranks (right). We find that stronger average alignment for the final stage leads to better performance. Similarly, larger rank leads to better loss and stronger task performance.

## 6 CONCLUSION

Training and inference are usually coupled, but Network of Theseus (NoT) breaks this coupling by progressively replacing a guide network with target modules while aligning intermediate representations; therefore, the trained function can be carried into a different inference architecture. Across ImageNet and Wikitext-103, this staged conversion preserves much of the guide's performance for large cross-family changes, specifically ResNet-18→MLP, DINOv2→Patch-MLP, ResNet-50→ResNet-18, GPT-2→RNN, whereas naive replacement collapses. Therefore, representational alignment is the key mechanism.

Regardless of whether the guide architecture was trained for a downstream task or not, we find that such staged replacement allows for an inductive bias transfer, allowing us to discover target architectures that preserve performance in comparison to the guide architecture. Our results raise concrete scientific questions: What defines a "discoverable representation", one that a different target architecture can reliably realize, and how can training objectives be designed to increase discoverability? Given NoT decouples training from deployment, how should we jointly choose guide architectures (for favorable optimization geometry) and target architectures (for deployment desiderata such as latency, memory, and parallelism) to trace the optimal alignment–efficiency frontier?

We believe NoT reframes how we think about architectural design. If representational alignment can be used to carry performance across network architecture families, then the new goal of an architecture during training is to construct discoverable representations rather than dictating the deployed inference architecture.

**Limitations**: We have not fully explored the space of architectures in our current work and, more importantly, have not reached the boundary of the performance of our alignment procedure for two reasons: we have not reached a point of overfitting in Figure 6 that increased target network capacity continues to improve performance. This lack of coverage is due to computational limitations as the progressive process requires more resources that simply fitting all layers at once. This gap between our performance and the possible best performance is driven by certain bottleneck layers, especially optimizing the last layers as shown in Figure 5. Further training methods can be explored to improve these results such as more rich data augmentation.

## ETHICS STATEMENT

The authors acknowledge that this work follows the Code of Ethics outlined by ICLR 2026. This paper uses publicly available data and pretrained models.

## REPRODUCIBILITY STATEMENT

We include all details of our algorithm and methodological choices in Appendix C. We also include learning rates, number of epochs of training, warmup epoch settings, learning rate scheduler usage in Appendix E. We include this across all stages for what we find to be the best performance. We use open-source models and datasets that are easily available and can be downloaded. We also explain our compute usage clearly. We include anonymous source code as part of the submission.

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

# A    APPENDIX OVERVIEW

> The ship wherein Theseus and the youth of Athens returned had thirty oars, and
> was preserved by the Athenians down even to the time of Demetrius Phalereus,
> for they took away the old planks as they decayed, putting in new and stronger
> timber in their place, insomuch that this ship became a standing example among
> the philosophers, for the logical question of things that grow; one side holding that
> the ship remained the same, and the other contending that it was not the same.
> –Plutarch

In Appendix B, we shortly cover LLM usage in this paper. We present additional details of NoT, experiments and analysis, as well as additional results. In Appendix C, we provide further details of NoT, providing an overview of the method and covering the similarity metrics applied in this paper, CKA and D-MNN in detail. In Appendix E, we provide additional details on our experiments, going into further detail on training and evaluation. We cover our batch normalization recalibration as well as general details on our ResNet-50→ResNet-18 result. In Appendix F, we recreate Figure 5 with D-MNN for further analysis with D-MNN. In Appendix G, we provide further analysis of DINOv2 to investigate why we can remove cross-token calculations.

# B    LLM USAGE

LLMs were used for editing and minor changes in this paper. LLMs were not used in any stage of this process. We did not generate any content with LLMs, mainly using them for grammatical correction or shortening. No major ideation or writing was done with any LLM.

# C    METHODS OVERVIEW

We give an overview of NoT in Algorithm 1 and cover more methodological details here.

---

**Algorithm 1 Network of Theseus (NoT):** staged representational alignment and conversion

---

**Require:** Guide $f^G(\cdot; \theta^G)$ (frozen), target $f^T$ (init as $f^G$), replacement mapping $\mathcal{R}$ (shape-compatible), schedule $\mathcal{S} = (I_1, \ldots, I_T)$ (default progressive), unlabeled inputs $\mathcal{X}$, labeled data $\mathcal{D} = \{(x, y)\}$, similarity $g$ (e.g., CKA); define $\Delta(\boldsymbol{A}, \boldsymbol{B}) = 1 - g(\boldsymbol{A}, \boldsymbol{B})$
1: **for** $t = 1$ to $T$ **do**                                                                    ▷ Alignment stage $t$
2:     # Replace layer $t$, assume previous layers have been replaced
3:     $J_t \leftarrow I_t \setminus I_{t-1}$; **replace** $\{i \in J_t\}$ in $f^T$ with $\{\mathcal{R}(i)\}$ (initialize)
4:     **freeze** $\theta^T_{\overline{I_t}}$ and **unfreeze** $\theta^T_{I_t}$                              ▷ no drift outside replaced set
5:     **for** mini-batches $x \sim \mathcal{X}$ **do**
6:         For each $i \in I_t$: $\boldsymbol{A}^G_i \leftarrow$ activations of layer $i$ in $f^G(x)$; $\boldsymbol{A}^T_i \leftarrow$ activations of layer $i$ in $f^T(x)$
7:         $L_{\text{align}} \leftarrow \Delta(\boldsymbol{A}^T_i, \boldsymbol{A}^G_i)$
8:         Update $\theta^T_{I_t}$ to minimize $L_{\text{align}}$
9:     **end for**
10: **end for**
11: **Fine-tune**: unfreeze all $\theta^T$ and minimize $L_{\text{task}}(f^T(x), y)$ over $(x, y) \in \mathcal{D}$
12: **return** converted target $f^T$

---

## C.1    CENTERED KERNEL ALIGNMENT

To compare representations, we use a representation similarity metric, $\mathcal{M}$, which corresponds to centered kernel alignment (CKA) (Kornblith et al., 2019; Cortes et al., 2012; Cristianini et al., 2001) in our setting. We specifically consider linear CKA.

CKA uses kernel functions on mean-centered representations to compute representational similarity matrices, which are then compared via the Hilbert-Schmidt Independence Criterion (HSIC). More specifically, suppose we have two sets of representations $\boldsymbol{A} \in \mathbb{R}^{b \times d_1}$ and $\boldsymbol{B} \in \mathbb{R}^{b \times d_2}$. We first compute the Gram matrices for each set of representations

$$K = AA^T, L = B'B'^T \tag{3}$$

We center the Gram matrices by introducing a matrix, $H$, where $H = I_b - \frac{1}{b}\mathbf{1}\mathbf{1}^T$.

$$\tilde{K} = HKH, \tilde{L} = HLH \tag{4}$$

We compute the HSIC on the Gram matrices.

$$HSIC(K, L) = \text{tr}(\tilde{K}, \tilde{L}) \tag{5}$$

Finally, we define our linear CKA metric as:

$$g_{\text{CKA}}(A, B') := \text{CKA}(K, L) = \frac{HSIC(K, L)}{\sqrt{HSIC(K, K) * HSIC(L, L)}} \tag{6}$$

In our setting, we consider representational *dissimilarity* and aim to minimize the dissimilarity between representations from our target network and guide network. We define this as:

$$\Delta_{\text{CKA}}(A, B) = 1 - g(A, B) \tag{7}$$

Linear CKA ranges from $0$ to $1$ (very different representations). Because of this, we take the complement by subtracting the linear CKA from 1 to represent dissimilarity. Most importantly, before computing $g$, rows of $A$ and $B$ are $\ell_2$-normalized.

### C.1.1 UNBIASED CKA

We use an unbiased estimator (Song et al., 2007) for HSIC for our GPT-2 Large→GPT-2 Small experiments, motivated by findings covered in Appendix D. We cover the unbiased HSIC here.

We start with our Gram matrices, $K, L$. An unbiased estimator for HSIC then involves forming zero-diagonal copies of both matrices.

$$\tilde{K} := K - \text{diag}(K), \qquad \tilde{L} := L - \text{diag}(L), \tag{8}$$

so that $\tilde{K}_{ii} = \tilde{L}_{ii} = 0$ and $\tilde{K}_{ij} = K_{ij}, \tilde{L}_{ij} = L_{ij}$ for $i \neq j$. Let $\mathbf{1} \in \mathbb{R}^m$ be the all-ones vector. The unbiased estimator of HSIC is

$$HSIC_U(K, L) = \frac{1}{m(m-3)}\left[\text{tr}(\tilde{K}\tilde{L}) + \frac{\text{tr}(\tilde{K}\tilde{L})}{(m-1)(m-2)} - \frac{2}{m-2}\text{tr}(\tilde{K}\tilde{L})\right], \tag{9}$$

which is well-defined for $m \geq 4$. The last term corresponds to the unbiased correction.

### C.2 DIFFERENTIABLE MUTUAL NEAREST NEIGHBORS

For Differentiable MNN (D-MNN), we use Huh et al. (2024) and Alshammari et al. (2025) to reformulate nearest neighbors into a differentiable form.

Let $b$ be the mini-batch size and anchors $i \in \{1, \ldots, b\}$. Two encoders $f_1, f_2$ produce features $\phi_i = f_1(x_i) \in \mathbb{R}^d$, $\psi_i = f_2(y_i) \in \mathbb{R}^d$. Define $\mathcal{J}_i = \{1, \ldots, b\} \setminus \{i\}$. For $k \in \mathbb{N}$, let $s_{f_1}(i), s_{f_2}(i) \subset \mathcal{J}_i$ be the index sets of the $k$ nearest neighbors of $i$ under $\{\phi_j\}$ and, $\{\psi_j\}$, respectively.

**k-Nearest Neighbors**: As defined by Huh et al. (2024), the overlap score for anchor $i$ is

$$g_{NN}(\phi_i, \psi_i) = \frac{|s_{f_1}(i), s_{f_2}(i)|}{k} \tag{10}$$

and the batch score is the average over $i$.

**From overlap to probabilities**: We introduce hard conditional distributions uniform on the kNN sets:

$$p_{f_1}(j \mid i) = \frac{\mathbf{1}\{j \in s_{f_1}(i)\}}{k}, \qquad q_{f_2}(j \mid i) = \frac{\mathbf{1}\{j \in s_{f_2}(i)\}}{k}, \quad j \in \mathcal{J}_i. \tag{11}$$

Then

$$\sum_{j \in \mathcal{J}_i} p_{f_1}(j \mid i) \, q_{f_2}(j \mid i) = \frac{|s_{f_1}(i) \cap s_{f_2}(i)|}{k^2}, \quad \Rightarrow \quad g_{NN}(\phi_i, \psi_i) = k \sum_{j \in \mathcal{J}_i} p_{f_1}(j \mid i) \, q_{f_2}(j \mid i). \tag{12}$$

**Differentiable conditional neighborhoods**: Let $\tau > 0$ be a temperature. Define pairwise similarities either by

$$s_{ij}^f = -\frac{\|\phi_i - \phi_j\|_2^2}{\tau} \quad \text{or} \quad s_{ij}^f = \frac{\phi_i^\top \phi_j}{\tau}, \qquad j \in \mathcal{J}_i, \tag{13}$$

and analogously $s_{ij}^g$ from $\{\psi_j\}$. Let $T_k^f(i) \subset \mathcal{J}_i$ and $T_k^g(i) \subset \mathcal{J}_i$ denote top-$k$ index selections under $s_{ij}^f$ and $s_{ij}^g$, respectively. We define masked, normalized conditionals over the top-$k$:

$$P_{ij} \equiv p_{f_1}(j \mid i) = \frac{\exp(s_{ij}^{f_1}) \, \mathbf{1}\{j \in T_k^{f_1}(i)\}}{\sum_{j' \in T_k^{f_1}(i)} \exp(s_{ij'}^{f_1})}, \qquad Q_{ij} \equiv q_{f_2}(j \mid i) = \frac{\exp(s_{ij}^{f_2}) \, \mathbf{1}\{j \in T_k^{f_2}(i)\}}{\sum_{j' \in T_k^{f_2}(i)} \exp(s_{ij'}^{f_2})}. \tag{14}$$

(Any differentiable soft-top-$k$ operator may be used; with an exact mask and $\tau \to 0$, $P_{ij}$ and $Q_{ij}$ converge to the hard uniforms.)

**D-MNN alignment and limit**: We define the differentiable alignment for anchor $i$ as the inner product

$$g_{\text{soft}}(\phi_i, \psi_i) = \sum_{j \in \mathcal{J}_i} P_{ij} \, Q_{ij}, \qquad \bar{g}_{\text{soft}} = \frac{1}{b} \sum_{i=1}^b m_{\text{soft}}(\phi_i, \psi_i). \tag{15}$$

Under the exact top-$k$ mask and $\tau \to 0$, $P_{ij}, Q_{ij} \to \frac{1}{k}\mathbf{1}\{j \in s_{f_1}(i)\}, \frac{1}{k}\mathbf{1}\{j \in s_{f_2}(i)\}$, so that $k \, g_{\text{soft}}(\phi_i, \psi_i) \to g_{NN}(\phi_i, \psi_i)$ by equation 12.

**Training objective**: Excluding the anchor, we write $P_{i\backslash *} = \{P_{ij}\}_{j \in \mathcal{J}_i}$ and similarly for $Q$. When $P$ (from $f$) guides $Q$ (from $g$), we minimize

$$\Delta_{\text{D-MNN}} := \mathcal{L}_{KL} = \frac{1}{b} \sum_{i=1}^b D_{KL}\Big(P_{i\backslash *} \,\Big\|\, Q_{i\backslash *}\Big). \tag{16}$$

This KL aligns conditional neighborhood distributions. Most importantly, before computing $g$, rows of $\boldsymbol{A}$ and $\boldsymbol{B}$ are $\ell_2$-normalized. For our setting in this paper, we have to tune the temperature for D-MNN.

### C.3 METHODOLOGY LIMITATIONS

NoT has a number of limitations as covered previously. We go deeper into methodological limitations here. NoT can be expensive memory-wise and runtime-wise. Each stage of NoT requires independent tuning. We also find that training can take a significant amount of time. For some conversions, we ran some conversions for up to 50 epochs of training for the best conversion. We believe these limitation may be overcome with further training optimizations and techniques such as more rich data augmentation and stronger label smoothing as used in previous papers (Bachmann et al., 2023).

## D  BIASED VS UNBIASED CKA: GPT-2 LARGE→GPT-2 SMALL

In our GPT-2 Large→GPT-2 Small setting, we found that using biased CKA as our alignment metric between GPT-2 Small transformer blocks and three GPT-2 Large transformer blocks did not have significant improvement. This is explained by inflated biased CKA values between GPT-2 Large blocks and GPT-2 Small blocks. We found that starting CKA values between randomly initialized

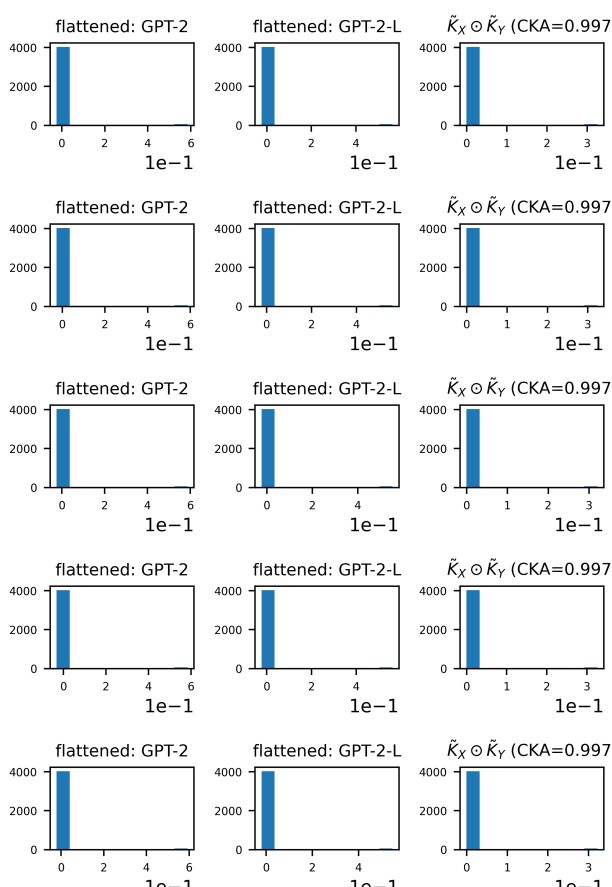

Figure 7: **Biased CKA inflated similarity scores between GPT-2 Small and GPT-2 Large.** When plotting values in the Gram matrices of a random pair of layers from GPT-2 Small and GPT-2 Large, we find that the Gram matrix values are very skewed. This is lead by inflated diagonal scores and skewed Gram matrix comparisons. We fix this using an unbiased HSIC estimator.

GPT-2 Small blocks and trained GPT-2 Large blocks started at 0.98, leading to very small optimization gains from optimizing CKA and very small improvements on downstream performance with Wikitext-103.

To better understand this trend, we plotted the distribution of values in our Gram matrices as a histogram in Figure 7. These plots correspond with visualizing the Gram matrices for 5 batches of 256 examples. The first two columns correspond to a Gram matrix as described by Equation (3) and the third columns corresponds to the product across the Gram matrices of the two models used by HSIC. We can see that the Gram matrices are heavily skewed with some values appearing to be much higher. These values correspond with the self-similarity in the networks, leading to inflated CKA scores. To mitigate this, we removed computing the HSIC with the diagonal using unbiased CKA (Song et al., 2007), which we discussed in Appendix C.1.1.

We applied UCKA in other experimental settings but found either degraded or had comparable results to using CKA.

## E  EXPERIMENTAL DETAILS

All training experiments were done on 8 H100 GPUs over 2 months in total. GPU optimization techniques were taken such as gradient accumulation and gradient checkpointing and some experiments with converting ResNet-18 to an MLP used mixed-precision training.

In Table 4, we show a table of learning rates used at each stage of training for our replacements. We also show our task training details in Table 5.

### E.1  HANDLING BATCHNORM

A difficulty with NoT is handling networks which use batch normalization, a common design choice in convolutional networks. However, in NoT, we find that batch normalization requires additional design considerations to ensure proper architectural transfer. For example, batch-feature statistics may shift for the replaced layers, meaning that batch normalization may need changes. We find that recalibrating batch normalization during representational similarity optimization has adverse effects on downstream performance.

To mitigate this, we freeze batch normalization calculations in the guide network during replacement. This also makes the replacement more difficult for the target modules. After representational similarity optimization, we recalibrate the batch normalization layers in the resultant target architecture before task training. The recalibration only focuses on tuning the batch normalization layers before doing target training. We find that this separation is useful, changing performance by a significant amount.

### E.2  RESNET-50 REPLACEMENT

Beyond direct layer/block replacements, we generalize NoT across architectures to convert from one architecture to another more generally. To do this, we relax the constraint requiring replaced components of our guide network to functionally fit into the guide network i.e. the replaced target module must accept the correct input and return the correct output. Having this constraint was necessary for Figure 4 to train different hybrid networks as we replace layers in the guide network.

See Figure 3. ResNet-50 blocks can be replaced with ResNet-18 blocks without incorporation into the full architecture. A full hybrid architecture with ResNet-18 and ResNet-50 blocks is unnecessary in this setting since the main goal is to reconstruct ResNet-18 while optimizing representational similarity with ResNet-50. Our findings shown in Table 1 and Table 3 are exciting because it demonstrates that we can make conversion across architectures general. For example, in Figure 3, ResNet-50 can be replaced with any architecture like GPT-2, and this can be converted to a smaller transformer or a smaller RNN.

| Model | Stage | Trained Learning Rate | Untrained Learning Rate | D-MNN Learning Rate | Number of Epochs |
|---|---|---|---|---|---|
| ResNet-18→MLP | 1 | $2 \times 10^{-5}$ | $2 \times 10^{-5}$ | $1 \times 10^{-5}$ | 5 |
| | 2 | $2 \times 10^{-5}$ | $2 \times 10^{-5}$ | $1 \times 10^{-5}$ | 30 |
| | 3 | $2 \times 10^{-5}$ | $2 \times 10^{-5}$ | $7.5 \times 10^{-6}$ | 25 |
| | 4 | $2 \times 10^{-5}$ | $2 \times 10^{-5}$ | $7.5 \times 10^{-6}$ | 25 |
| | 5 | $2 \times 10^{-5}$ | $2 \times 10^{-5}$ | $7.5 \times 10^{-6}$ | 30 |
| | 6 | $1.5 \times 10^{-5}$ | $1.5 \times 10^{-5}$ | $7.5 \times 10^{-6}$ | 30 |
| | 7 | $1.5 \times 10^{-5}$ | $1.5 \times 10^{-5}$ | $7.5 \times 10^{-6}$ | 30 |
| | 8 | $1 \times 10^{-5}$ | $1 \times 10^{-5}$ | $5 \times 10^{-6}$ | 30 |
| | 9 | $1 \times 10^{-5}$ | $1 \times 10^{-5}$ | $5 \times 10^{-6}$ | 30 |
| | 10 | $1 \times 10^{-5}$ | $1 \times 10^{-5}$ | $5 \times 10^{-6}$ | 30 |
| | 11 | $1 \times 10^{-5}$ | $1 \times 10^{-5}$ | $5 \times 10^{-6}$ | 30 |
| | 12 | $7.5 \times 10^{-6}$ | $7.5 \times 10^{-6}$ | $5 \times 10^{-6}$ | 30 |
| | 13 | $7.5 \times 10^{-6}$ | $7.5 \times 10^{-6}$ | $5 \times 10^{-6}$ | 35 |
| | 14 | $5 \times 10^{-6}$ | $5 \times 10^{-6}$ | $1 \times 10^{-6}$ | 35 |
| | 15 | $2.5 \times 10^{-6}$ | $2.5 \times 10^{-6}$ | $1 \times 10^{-6}$ | 40 |
| | 16 | $1 \times 10^{-6}$ | $1 \times 10^{-6}$ | $1 \times 10^{-6}$ | 40 |
| | 17 | $1 \times 10^{-6}$ | $1 \times 10^{-6}$ | $1 \times 10^{-6}$ | 45 |
| | 18 | $1 \times 10^{-6}$ | $1 \times 10^{-6}$ | $1 \times 10^{-6}$ | 50 |
| | 19 | $1 \times 10^{-6}$ | $1 \times 10^{-6}$ | $1 \times 10^{-6}$ | 50 |
| | 20 | $1 \times 10^{-6}$ | $1 \times 10^{-6}$ | $1 \times 10^{-6}$ | 100 |
| DINOv2→Patch-MLP | 1 | $1 \times 10^{-4}$ | $1 \times 10^{-4}$ | N/A | 10 |
| | 2 | $1 \times 10^{-4}$ | $1 \times 10^{-4}$ | N/A | 35 |
| | 3 | $1 \times 10^{-4}$ | $1 \times 10^{-4}$ | N/A | 35 |
| | 4 | $1 \times 10^{-4}$ | $1 \times 10^{-4}$ | N/A | 35 |
| | 5 | $1 \times 10^{-4}$ | $1 \times 10^{-4}$ | N/A | 50 |
| | 6 | $1 \times 10^{-4}$ | $1 \times 10^{-4}$ | N/A | 50 |
| | 7 | $1 \times 10^{-4}$ | $1 \times 10^{-4}$ | N/A | 50 |
| | 8 | $1 \times 10^{-4}$ | $1 \times 10^{-4}$ | N/A | 50 |
| | 9 | $1 \times 10^{-4}$ | $1 \times 10^{-4}$ | N/A | 50 |
| | 10 | $1 \times 10^{-4}$ | $1 \times 10^{-4}$ | N/A | 50 |
| | 11 | $1 \times 10^{-4}$ | $1 \times 10^{-4}$ | N/A | 50 |
| | 12 | $1 \times 10^{-4}$ | $1 \times 10^{-4}$ | N/A | 100 |
| GPT-2→RNN | 1 | $1 \times 10^{-4}$ | $1 \times 10^{-4}$ | N/A | 10 |
| | 2 | $1 \times 10^{-4}$ | $1 \times 10^{-4}$ | N/A | 35 |
| | 3 | $1 \times 10^{-4}$ | $1 \times 10^{-4}$ | N/A | 35 |
| | 4 | $1 \times 10^{-4}$ | $1 \times 10^{-4}$ | N/A | 35 |
| | 5 | $1 \times 10^{-4}$ | $1 \times 10^{-4}$ | N/A | 50 |
| | 6 | $1 \times 10^{-4}$ | $1 \times 10^{-4}$ | N/A | 50 |
| | 7 | $1 \times 10^{-4}$ | $1 \times 10^{-4}$ | N/A | 50 |
| | 8 | $1 \times 10^{-4}$ | $1 \times 10^{-4}$ | N/A | 50 |
| | 9 | $1 \times 10^{-4}$ | $1 \times 10^{-4}$ | N/A | 50 |
| | 10 | $1 \times 10^{-4}$ | $1 \times 10^{-4}$ | N/A | 50 |
| | 11 | $1 \times 10^{-4}$ | $1 \times 10^{-4}$ | N/A | 50 |
| | 12 | $1 \times 10^{-4}$ | $1 \times 10^{-4}$ | N/A | 100 |
| ResNet-50→ResNet-18 | 1 | $1 \times 10^{-3}$ | $1 \times 10^{-3}$ | N/A | 5 |
| | 2 | $1 \times 10^{-3}$ | $1 \times 10^{-3}$ | N/A | 15 |
| | 3 | $1 \times 10^{-3}$ | $1 \times 10^{-3}$ | N/A | 25 |
| | 4 | $1 \times 10^{-3}$ | $1 \times 10^{-3}$ | N/A | 35 |
| | 5 | $1 \times 10^{-3}$ | $1 \times 10^{-3}$ | N/A | 35 |
| | 6 | $1 \times 10^{-3}$ | $1 \times 10^{-3}$ | N/A | 35 |
| | 7 | $1 \times 10^{-3}$ | $1 \times 10^{-3}$ | N/A | 35 |
| | 8 | $1 \times 10^{-3}$ | $1 \times 10^{-3}$ | N/A | 100 |

Table 4: **Learning rates and epochs of training across NoT representational similarity optimization**: We report learning rates and number of epochs of training used for training our model replacements for NoT representational similarity matching. We find that different stages can have significantly different learning rate requirements.

# F   D-MNN STAGED ALIGNMENT

We recreate Figure 5 for our new metric, D-MNN (Appendix C.2) to analyze loss dynamics. We show results in Figure 8. We find that the quantitative loss for D-MNN is much higher and farther from 0, indicating little saturation. This is likely because D-MNN prioritizes locality over global geometry when comparing representations. We believe this demonstrates the benefit of our method:

# G   ANALYZING DINOv2

A surprising result we found was that we could replace attention in DINOv2 with a Patch-MLP. This Patch-MLP has no cross-token communication, with linear layers applying transformations

| Model | Learning Rate | Epochs | Warm-up | Scheduler | Notes |
|---|---|---|---|---|---|
| ResNet-18→MLP | $5 \times 10^{-4}$ | 100 | 10 | Linear+Cosine | Gradient Clipping to 1.0 |
| DINOv2→Patch-MLP | $1 \times 10^{-4}$ | 100 | 5 | Linear+Cosine | N/A |
| ResNet-50→ResNet-18 | $1 \times 10^{-3}$ | 100 | 0 | None | N/A |
| GPT-2→RNN | $1 \times 10^{-4}$ | 100 | 5 | Linear+Cosine | Gradient Clipping to 1.0 |

Table 5: **Learning rates, epochs, and scheduler settings for NoT task training.** We report learning rates, number of epochs of training, number of warmup epochs, and scheduler details for our settings along with miscallaneous details like using gradient clipping.

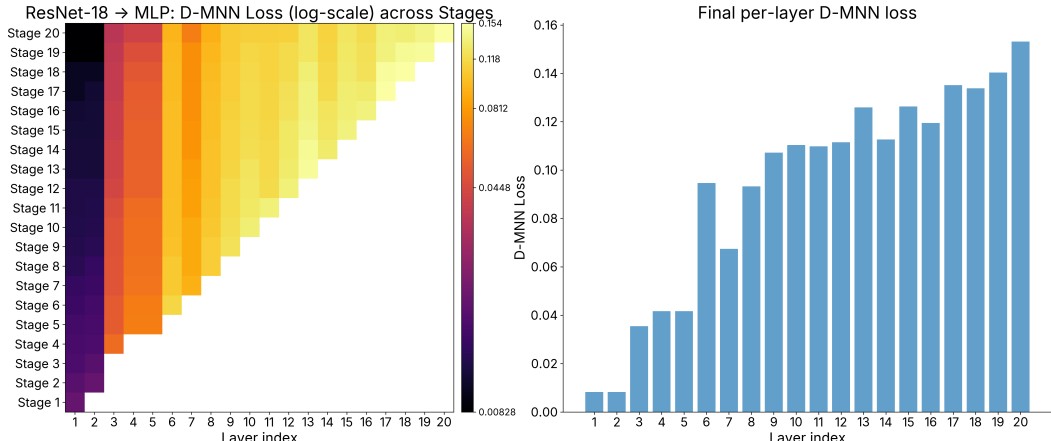

Figure 8: **D-MNN finds similar layer bottlenecks as CKA.** We show the per-stage losses for D-MNN, as with CKA. We find that D-MNN is far from saturating unlike CKA, which had loss values closer to 0. We find that D-MNN is also more consistent across layers, indicating that further optimization can be pushed for better alignment in D-MNN. We believe this shows even better correlation between D-MNN and final task accuracy and optimizing D-MNN further may lead to a task accuracy closer to the guide network.

per-token instead. We investigated why we could convert from attention to Patch-MLPs in DINOV2 without much loss in performance. First, we found that the first layer of DINOv2 is a convolution and the last layer aggregates over tokens for a label prediction, meaning that cross-token prediction is not entirely eliminated.

In Figure 9, we show attention map visualizations for all layers in DINOv2 for a set of ImageNet images. We see that for most DINOv2 layers, attention is highly sparse. Image tokens only focus on tokens in a nearby neighborhood. Most of the other focus goes to the original token itself. Cross-token communication is sparse. We believe this makes attention is DINOv2 much easier to replace with a Patch-MLP.

This matches intuition in prior findings that attention matrices in ViTs are repetitive across layers (Zhang et al., 2024c; Venkataramanan et al., 2023) when analyzing attention patterns when completing tasks like image classificaiton. Intuitively, this makes sense. In image classification, there is a central image and cross-token communication may not be entirely necessary. We are not the first to propose removing cross-token computation in ViTs. Other works such as MetaFormer (Yu et al., 2023) or RIFormer (Wang et al., 2023) have proposed methods to replace cross-token mixing in ViTs or MLP-Mixers (Tolstikhin et al., 2021) due to sparse cross-token communication.

We believe such findings demonstrate the benefits of NoT. We were able to match findings in prior work by directly training a replacement for attention with strong performance. This is indicative that image classification is sparse, requiring a network to focus on a specific token. However, we also believe that such findings may not transfer to other tasks. For example, if we were to focus on semantic segmentation, then we may find that we cannot use NoT for strong performance given the likelihood that semantic segmentation requires cross-token communication.

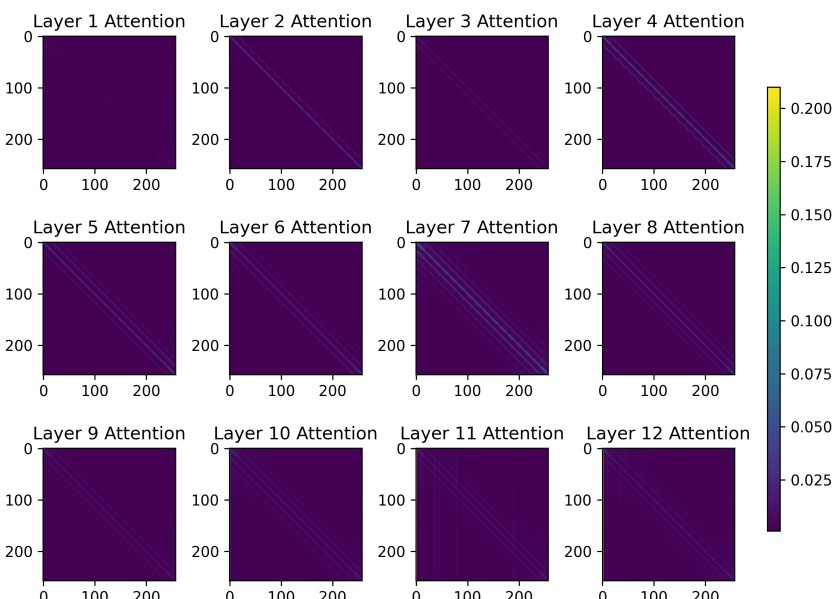

Figure 9: **DINOv2 has simple attention maps**: We visualize attention maps for DINOv2 over a set of ImageNet images to better understand how we are able to remove cross-token communication via NoT. We find attention is fairly sparse DINOv2, when focusing on image classification.

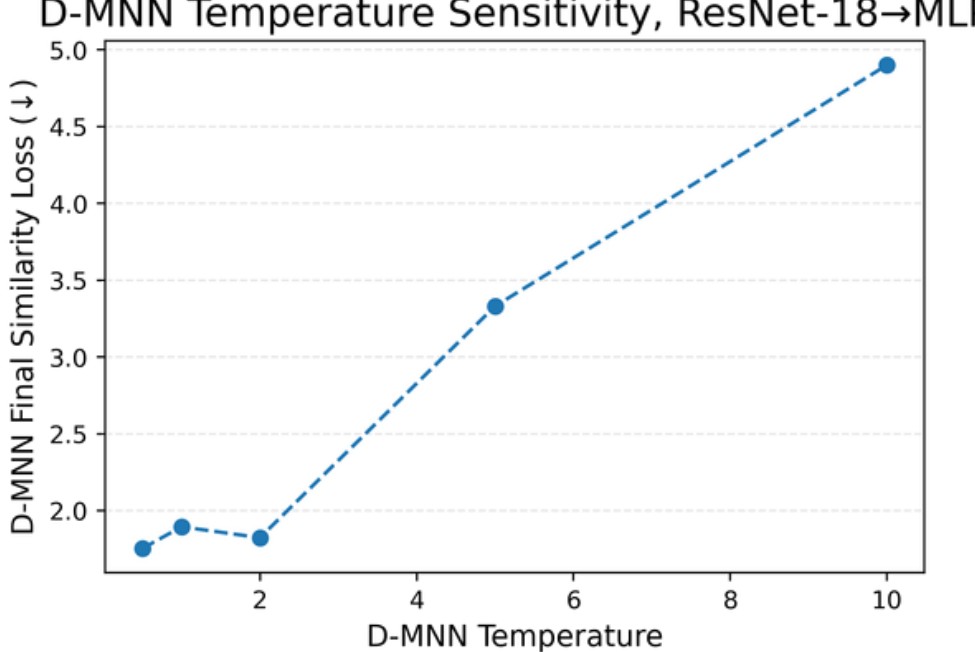

Figure 10: **Temperature Variations across D-MNN**: We analyze how D-MNN similarity loss varies across different temperature values from 0.5 to 10.0. We see that close temperature values like 0.5 to 2.0 have similarity loss. As we increase similarity we see worse similarity.

## H   D-MNN ANALYSIS

We analyze our new similarity metric, D-MNN in Figure 10, starting with analyzing how temperature in Equation (13) affects our final similarity loss for our ResNet-18→MLP conversion. We find that close temperature values have similar final similarity losses but as we make larger changes in temperature, we see lower D-MNN similarity between our original and converted network. This is

| Setting | Guide | NoT Target | Distillation Target | Original |
|---|---|---|---|---|
| ResNet-18→MLP | 69.16 | 62.55 | 59.13 | 33.36 |
| Untrained ResNet-18→MLP | 0.10 | 60.28 | 31.11 | 33.36 |
| GPT-2→RNN (↓) | 37.50 | 50.58 | 75.49 | 121.19 |
| Untrained GPT-2→RNN (↓) | 51948.4 | 58.26 | 135.55 | 121.19 |

Table 6: **Progressive distillation**: We include a comparison to knowledge distillation as a method for aligning our converted architectures to our original architectures. We progressively convert and distill ResNet-18 into our MLP and GPT-2 into our RNN. We only apply this method for conversion where the guide and target have the same layer shapes to allow propogation to the final logits. We find competitive results with NoT when the guide is trained and much worse results when the guide is untrained. This emphasizes that NoT can exploit untrained networks.

| | Original | NoT | Baseline |
|---|---|---|---|
| ConvNeXt→Patch-MLP | 82.22 | 64.14 | 58.43 |

Table 7: **ConvNeXt has similar performance under NoT.** We conduct a conversion from every convolutional layer in ConvNeXt to a linear layer to achieve a different Patch-MLP from DINOv2. We find that we preserve performance better than when we train the network from scratch.

intuitive due to our design of D-MNN, where larger temperatures flatten the probability distribution of the target and guide network representations.

# I COMPARISON WITH PROGRESSIVE DISTILLATION

While in this work, we main use representational methods to align our converted target architecture to the guide architecture, another method to perform such alignment is knowledge distillation (Hinton, 2015). We introduce knowledge distillation as part of our progressive replacement schedule as a comparison to NoT, which we refer to as *progressive distillation*.

At each replacement stage, we convert a guide network layer to a target layer and performance knowledge distillation from the guide network to the newly converted architecture. We train the converted layers using the distillation. Due to needing to pass activations to the final logit layer, we only test settings where the activation shapes are the same across the converted layers in the guide network.

We show results in Table 6. We find that progressive distillation performs competitively with NoT with CKA when the guide architecture is trained but performs worse than the target architecture baseline when the guide architecture is untrained. This is intuitive because distillation doesn't provide access to the internal geometry of the architecture which we exploit with CKA. This further establishes that using representational similarity metrics explains our improvement with untrained architectures.

One interesting finding we would like to highlight is that we find a significant improvement with NoT in our GPT-2→RNN setting than over distillation. This implies that NoT overcomes a training problem with RNNs that could not be overcome with distillation. Using the output of a teacher model was not sufficient to prevent stability problems in training an Elman RNN. However, using internal representations of a guide architecture could prevent said problems and achieve stronger results. We believe this has significant implications for RNN training and demonstrates that RNN training stability can be improved by using untrained architectures.

# J ADDITIONAL RESULTS: CONVNEXT

We add additional results with ConvNeXt (Liu et al., 2022) in Table 7. In ConvNeXt, 1x1 convolutions are applied to the internal representations to mix tokens. Similar to DINOv2, we can replace the attention mechanism with a patch-wise MLP that prevents interactions across tokens, only allowing the model to use the representations from the original architecture.

|  | Original | NoT (Trained-EMA) | NoT (Trained-Original) |
|---|---|---|---|
| GPT-2→RNN | 37.50 | 48.60 | 50.58 |

Table 8: **Automating NoT with EMA and new learning rate schedulers**: We aim to automate NoT by introducing several extensions that automatically tune the learning

We find that using NoT allows for improved performance over training from scratch. This demonstrates that NoT can be applied to newer architectures with different designs with minimal loss inf performance. We leave further exploration of different architectural modifications as future work.

## K    AUTOMATING NOT

One major component we aim to address is that NoT seems to rely on a hand-designed schedule where every layer has the learning rate and number of epochs of training tuned for optimal representational similarity. To overcome this limitation, we also discuss further methodology for automating the learning rate and number of epochs of training applied per stage of representational similarity optimization.

Our automation applies an exponential moving average (EMA) over the representational similarity loss at every stage to measure the average change over a 500 step window. We then use this EMA as a learning rate condition and stopping condition. For our learning rate, we start with a consistent setting for all stages. We reduce the learning rate when our EMA plateaus over 4 epochs (PyTorch Contributors, 2025). We then stop our optimization when we saturate the EMA of the representational similarity loss i.e. we reach a CKA loss of less than 0.01 or train for 100 epochs. We find that this threshold is sufficient to get a strong result.

We apply this to our GPT-2→RNN result in Table 8. We convert GPT-2 to an RNN with a starting learning rate of 1e-4 at every stage, allowing the scheduler to tune the learning rate, and using an EMA over 500 steps to measure when to apply the scheduler and when to stop the training. For each stage, we either train until the similarity loss for CKA is less than 0.01 or stop training after 100 epochs. We find that our automatic tuning leads to a slightly stronger result than what was reported for NoT. This reduces the expensive nature of NoT.

