# OpenReview forum: "Network of Theseus (like the ship)"
_ICLR.cc/2026/Conference — Submitted to ICLR 2026_

### Official Review · Reviewer_5FPe · 2025-10-28

**Soundness:** 3
**Presentation:** 2
**Contribution:** 2
**Rating:** 4
**Confidence:** 4

**Summary:**

This paper proposes a framework for gradually replacing parts of a neural network. It is based on the Ship of Theseus philosophical question about, how if we gradually replace parts of a vessel, will it eventually become a new vessel altogether?

The paper considers a similarity loss to match internal representations of data at every stage. Experiments are conducted on computer vision and natural language processing tasks.

**Strengths:**

- The idea of the paper is quite interesting as a way to reframe how architecture training and inference is considered.
- The idea combines multiple prior works like knowledge distillation.
- Application is not restricted to one task domain such as computer vision.

**Weaknesses:**

- The paper distinguishes itself from compression pipelines, such as pruning and quantization. However, those pipelines are utilized to maintain end-to-end performance while achieving some hardware benefit, such as lower latency or power consumption. However, little emphasis is placed on such gains in the experimental section of this work, if at all, which severely limits its utility and merit.
- The overall idea of the paper seems incremental and generally just based on loss on matching hidden representations, which prior works for knowledge distillation [1] have done.
- The paper should compare to prior work which using gradual techniques to shrink an architecture [2] by phasing out some components.
- Experimental results are largely confined to older/smaller architectures, limiting the scope of utility of this work.

**Questions:**

See weaknesses.

References:

[1] https://arxiv.org/abs/1910.01108

[2] https://arxiv.org/abs/2305.12972

---

> ### Author Response · Authors · 2025-11-20
> **Response 1/2**
>
> We thank the reviewer for their thoughtful review. We are glad the reviewer found our paper “interesting” and appreciated that the application is not restricted to “one target domain”. We hope to address concerns here.
>
> > W1: However, little emphasis is placed on such gains in the experimental section of this work, if at all, which severely limits its utility and merit.
>
> Thank you for the thoughtful comment. Our goal in this paper is cross‑architecture transfer: using representational alignment to carry a trained or untrained guide’s internal computations into a different target architecture. This decouples training from deployment and opens design points that traditional compression pipelines cannot reach. These choices are intentionally operator‑level simplifications, for example, removing quadratic token mixing or reducing depth, that are widely associated with lower memory traffic and more hardware‑friendly kernels. Our specific contribution is the conversion method rather than system benchmarking because system‑level performance improvements are highly platform‑dependent. To keep the paper focused, we isolate the algorithmic question: can we preserve performance while changing the architecture itself? We show this across different conversions where the targets are, by design, simpler to implement or run. In particular, our DINOv2→Patch‑MLP analysis shows attention maps are sparse and largely local, explaining why removing cross‑token mixing keeps accuracy while simplifying computation. This points to practical efficiency upside even without reporting device‑specific numbers.
>
> Importantly, NoT is complementary to pruning/quantization rather than a replacement. NoT first moves you to an inference architecture that better matches deployment constraints. Standard compression can then be applied on top of that target. In this sense, our experiments establish the precondition for utility while the exact latency/power gains depend on the chosen hardware stack and can be pursued in system‑oriented follow‑ups.
>
> > W2: which prior works for knowledge distillation [1] have done
>
> We compare knowledge distillation and show results here as well as in Appendix Section I. We believe progressive knowledge distillation falls under the scope of approaches in our paper since the method involves staged replacement with an alignment metric, which is aligning output logits. To the best of our knowledge, distillation has not been applied across radical architectural shifts.
>
>  |    | Guide | NoT Target | Distillation Target | Original |
> |-------------------------------------|-------|------------|---------------------|----------|
> | ResNet-18$\rightarrow$MLP           | 69.16 | 62.55      | 59.13    | 33.36    |
> | Untrained ResNet-18$\rightarrow$MLP | 0.10  | 60.28      | 31.11    | 33.36    |
> | GPT-2$\rightarrow$RNN ($\downarrow$)                | 37.50 | 50.58      | 75.49               | 121.19   |
> | Untrained GPT-2$\rightarrow$RNN ($\downarrow$)     |   51914.5    | 58.26      | 135.55              | 121.19   |
>
> As expected, we improve significantly when using a trained model as our teacher than when using an untrained model, demonstrating the uniqueness of our approach. We will incorporate this into the main paper and include it upon acceptance.
>
>
> > W3: The paper should compare to prior work which using gradual techniques to shrink an architecture [2] by phasing out some components.
>
> Thanks for the reference. If we understand the reviewer correctly, they mean that we can use Equation (1) in the paper to gradually convert layers to identity functions in order to shrink the given architecture. Equation (1) defines an activation annealing scheme with a $\lambda$ penalty that increases during training so that the activation becomes an identity and two adjacent convolutions can be merged for inference. This is a within‑architecture training method to prune non‑linearities and fuse layers. It neither aligns internal states with a separate model nor changes operator families.
>
> By contrast, NoT performs cross‑architecture conversion by stage‑wise representational alignment to a fixed guide. Each replaced module is optimized to match the guide’s intermediate activations, often with different computations and shapes, using unlabeled data and even untrained guides, before a final task fine‑tune. Turning layers into identities cannot realize these operator changes or the guide‑target alignment that NoT required, so that equation does not apply to our setting. We will clarify this distinction and cite VanillaNet in the related‑work section.

---

> > ### Author Response · Authors · 2025-11-20
> > **Response 2/2**
> >
> > > W4: older architectures
> >
> > Thank you for the feedback. We would like to point out that we do use DINOv2 [1] as part of our comparison as well, which is a relatively newer architecture. Furthermore, newer architectures come with significant compute cost, far beyond what we can run with academic-level compute. From our view, the use of older architectures indicates a stronger likelihood that our methods can transfer to newer architectures. Based on this feedback, we are also currently including a conversion from ConvNeXt$\rightarrow$Patch-MLP as well to include newer architectures.
> >
> > [1] Oquab et. al. DINOv2: Learning Robust Visual Features without Supervision. TMLR, 2023.

---

> > > ### Comment · Area_Chair_pm6X · 2025-11-26
> > >
> > > Dear reviewer 5FPe,
> > >
> > > Could you take a look at the authors response and leave your feedback.
> > >
> > > AC

---

> > > ### Comment · Reviewer_5FPe · 2025-11-26
> > > **Maintaining score**
> > >
> > > The reviewer thanks the authors for their efforts.
> > > After carefully reading the author rebuttal, I maintain concurrence with the other reviewers that this paper falls short of the acceptance threshold and thus elect to maintain my score.

---

> ### Author Response · Authors · 2025-11-29
>
> Thank you for your response. We would also like to add that we had finished an experiment with ConvNeXt [1]. We convert every 1x1 convolutional layer in ConvNeXt into a Linear layer which prevents patch-wise mixing.
>
> We report results below for this additional experiment and add this to Appendix Section J as well.
>
> | Experiment             | Original | NoT   | Baseline |
> |------------------------|----------|-------|----------|
> | ConvNeXt --> Patch MLP | 82.22    | 64.14 | 58.43    |
>
>
> Every experiment we have added has supported our finding that stage-wise representational similarity is useful for converting across architectures. This additional experiment emphasizes the generality of NoT even further. If the reviewer has specific concerns, we would be happy to address them and discuss further.
>
> Overall, we would like to emphasize our effort to address comments from the reviewer. We have added numerous experiments like newer architectures for NoT with GPT-2 Large and ConvNeXt and clarified our novelty in NoT in comparison to other work. Requested baselines like progressive distillation were also added. We hope that this can be taken into consideration now that reviewer responses have been muted.
>
> [1] Liu et. al. A convnet for the 2020s. CVPR 2022.

---

### Official Review · Reviewer_zw5V · 2025-10-31

**Soundness:** 3
**Presentation:** 3
**Contribution:** 2
**Rating:** 4
**Confidence:** 2

**Summary:**

This paper proposes a 2-step method that given an initial guide network produces a desired target network. In the first step, components of the guide network (such as layers or blocks) are iteratively replaced one at a time with target components via optimisation of a metric between activations to produce a desired target network. In step 2 the target network is trained end-to-end on the downstream task. The paper proposed use of two different metrics.
Experiments were performed with ImageNet for an image classification task, and Wikitext for a language modelling task. Firstly, the method was shown to perform better than training the target network from scratch. Then progressive replacement was shown to perform better than 3 other replacement schedules. Finally it was shown that the method retained performance improvement over training from scratch even when the guide was untrained.

**Strengths:**

The method was clearly explained and the experiments were relevant.

**Weaknesses:**

The paper could more concisely summarise its contributions in the introduction. It was clear what the method did but less clear what was novel.

The paper would be improved by further comparison to other methods.

The paper says “Doing the same with distillation would lead to much worse results.”. Why isn’t this just demonstrated empirically in a direct comparison with the proposed method?
The results only compare the guide accuracy, NoT accuracy, and from-scratch baseline accuracy. Why was the method not compared to other knowledge distillation methods such as for example Task-aware layEr-wise Distillation (TED) proposed in Liang et al Less is More: Task-aware Layer-wise Distillation for Language Model Compression (ICML 2023)

The authors says “the stage-wise replacement of NoT has not been proposed in previous work”.
Xu et al. BERT-of-Theseus: Compressing BERT by Progressive Module Replacing (EMNLP 2020) uses a strategy of progressive replacement (while also using the Theseus analogy) where at each training step t, a guide module is replaced with the target with probability p_t. The idea appears similar to the current work but with a different replacement schedule. Why was this strategy not compared to?

**Questions:**

Did you try any other X -> MLP combinations other than ResNet-18 → MLP? Do guides that themselves can be trained from scratch to high accuracy act as better guides than networks that train from scratch to a lower accuracy?  Does an untrained MLP guide (MLP -> MLP)  used in NoT work poorly empirically?

The paper says "For all training, we use a consistent batch size of 256. Representational similarity metrics are affected by the batch size, specifically more samples allows the metric to better approximate similarity.". To what extent do the results for your method change with batch size? An ablation showing what happens if the batch size was halved and doubled would be helpful.

The paper says “We distinguish NoT from distillation. NoT aligns computations and can transfer inductive biases from one architecture to another”. I did not understand this. In what way does NoT align computations and distillation does not? Could this be clarified please.

---

> ### Author Response · Authors · 2025-11-20
> **Response 1/3**
>
> We thank the reviewers for their thoughtful review. We are glad that the reviewer found the methods in the paper “well-explained” and found the experiments “relevant”. We hope to address concerns here.
>
> > W1: summarise its contributions in the introduction
>
> We agree. We have edited the introduction in a new version, with edits in blue. Please take a look; we hope that this makes our contributions clearer. We also incorporate these contributions here for convenience:
>
> * We introduce the Network of Theseus (NoT), a method to part-by-part convert from one architecture to another using representational alignment.
> * We demonstrate NoT across a wide variety of architectural conversions, such as converting convolutions in ResNet-18 to linear layers to create an MLP or converting attention in GPT-2 to RNN layers to create a Deep RNN. We find that across all conversions, we preserve performance of the original architecture despite dramatic architectural shifts.
> * Surprisingly, we find that we can perform the same conversion starting from an untrained network architecture, demonstrating that we can transfer inductive biases to our target networks.
> * We validate our findings using representational similarity metrics like CKA and introduce a new metric based on a differentiable version of mutual nearest neighbors, D-MNN, to further validate our findings as general and invariant to metric.
>
> > W2: The paper says “Doing the same with distillation would lead to much worse results.”. Why isn’t this just demonstrated empirically in a direct comparison with the proposed method? The results only compare the guide accuracy, NoT accuracy, and from-scratch baseline accuracy. Why was the method not compared to other knowledge distillation methods
>
> The reviewer is correct. Based on this comment and other requests from reviewers, we add a baseline comparison to knowledge distillation progressively for our conversion from ResNet-18 to an MLP and GPT-2 to an RNN. We believe progressive knowledge distillation falls under the scope of approaches in our paper since the method involves staged replacement with an alignment metric, which is aligning output logits. To the best of our knowledge, distillation has not been applied across radical architectural shifts.  We report the findings below and results in Appendix I in the paper.
>
>
>
> |       | Guide | NoT Target | Distillation Target | Original |
> |-------------------------------------|-------|------------|---------------------|----------|
> | ResNet-18$\rightarrow$MLP           | 69.16 | 62.55      | 59.13    | 33.36    |
> | Untrained ResNet-18$\rightarrow$MLP | 0.10  | 60.28      | 31.11    | 33.36    |
> | GPT-2$\rightarrow$RNN ($\downarrow$)                | 37.50 | 50.58      | 75.49               | 121.19   |
> | Untrained GPT-2$\rightarrow$RNN ($\downarrow$)      |  51914.5     | 58.26      | 135.55              | 121.19   |
>
> As expected, we see that knowledge distillation does not work from an untrained network. This distinguishes our findings from findings in previous works.
>
> > W3: Xu et al. BERT-of-Theseus: Compressing BERT by Progressive Module Replacing (EMNLP 2020) uses a strategy of progressive replacement
>
> Thank you for the reference. We have added it to our related work and added some clarifications. We would like to point out that BERT‑of‑Theseus [1] is more narrow in scope than our paper. We can enable any‑to‑any architecture transfer in a way that, to our knowledge, hasn’t been empirically shown before. The novelty of our approach also stems from incorporating representational similarity, which, according to our findings, enables us to exploit untrained architectures. We add benchmarks to a related comparison, distillation. However, we believe that BERT‑of‑Theseus supports our findings of the effectiveness of our approach for cross‑architecture transfer, and we have incorporated it into our related work as well. In addition, BERT‑of‑Theseus focuses on within‑BERT compression with a stochastic replacement schedule during supervised training, whereas NoT aligns internal representations stage‑by‑stage and is designed for cross‑family conversions using alignment. The settings are distinct.

---

> > ### Author Response · Authors · 2025-11-20
> > **Response 2/3**
> >
> > > Q1: Did you try any other X -> MLP combinations other than ResNet-18 → MLP? Do guides that themselves can be trained from scratch to high accuracy act as better guides than networks that train from scratch to a lower accuracy? Does an untrained MLP guide (MLP -> MLP) used in NoT work poorly empirically?
> >
> > We have run a quick experiment where we convert a 3-layer CNN trained on CIFAR-10 to an MLP. We report the performance here:
> >
> > | CNN Guide | NoT MLP | MLP   |
> > |---------------------|---------|-------|
> > | 51.23             | 49.32   | 41.01 |
> >
> > We find that we have similar improvements as found in the paper. This indicates that our results are not driven by some aspect of ResNet-18 such as residual connections.
> >
> > Thanks for the question on untrained MLP guide to MLP target. We actually find no change in the performance from the original MLP. We report the results below.
> >
> > | Untrained MLP Guide | NoT MLP | MLP   |
> > |---------------------|---------|-------|
> > | 0.0922              | 32.75   | 33.36 |
> >
> >
> > > Q2: The paper says "For all training, we use a consistent batch size of 256. Representational similarity metrics are affected by the batch size, specifically more samples allows the metric to better approximate similarity.". To what extent do the results for your method change with batch size? An ablation showing what happens if the batch size was halved and doubled would be helpful.
> >
> > Unfortunately, due to difficulties with compute limitations, we cannot increase our batch size beyond 256. But we can demonstrate what happens when the batch size is halved. We also cannot completely run progressive training due to time constraints.
> >
> > We compare our representational similarity for joint training (from Table 3) when our batch size is halved to 128 and lowered to 64 in comparison to 256. We report our final similarity loss.
> >
> > | Batch Size | CKA Similarity Loss |
> > |------------|---------------------|
> > | 64         | 0.2476              |
> > | 128        | 0.1758              |
> > | 256        | 0.0694              |

---

> > > ### Author Response · Authors · 2025-11-20
> > > **Response 3/3**
> > >
> > > > Q3: The paper says “We distinguish NoT from distillation. NoT aligns computations and can transfer inductive biases from one architecture to another”. I did not understand this. In what way does NoT align computations and distillation does not? Could this be clarified please.
> > >
> > > Sorry, this could be more clear. In NoT, each replaced module in the target is trained to match the internal activations of the corresponding module in the guide—stage‑by‑stage—using a relational similarity metric. This preserves the relational geometry of one embedding to another but does not strictly match the absolute vectors as distillation would..
> > > Classical distillation primarily matches the outputs (teacher logits/soft targets) of a trained teacher, optionally with auxiliary feature hints, as an added loss during task training. It does not require the student to reproduce the teacher’s intermediate computations at specific locations, and it generally depends on a trained teacher’s predictions. By contrast, NoT’s stage‑wise representational alignment lets us transfer inductive biases even from untrained guides and perform any‑to‑any architectural conversions, because the constraint is on internal states, not on final task outputs.
> > >
> > > [1] Xu et al. BERT-of-Theseus: Compressing BERT by Progressive Module Replacing. EMNLP, 2020

---

> > > > ### Comment · Area_Chair_pm6X · 2025-11-26
> > > >
> > > > Dear reviewer zw5V,
> > > >
> > > > Could you take a look at the authors response and leave your feedback.
> > > >
> > > > AC

---

### Official Review · Reviewer_fJRw · 2025-11-01

**Soundness:** 3
**Presentation:** 3
**Contribution:** 3
**Rating:** 4
**Confidence:** 4

**Summary:**

This paper proposes Network of Theseus (NoT), aiming to relax the constraint induced by inductive bias that a neural network must share the same architecture during training and inference. NoT progressively replaces parts of a trained or even untrained “guide” network with heterogeneous target modules (e.g., CNN $\rightarrow$ MLP, GPT-2$\rightarrow$RNN) while maintaining similar representations measured by metrics such as CKA or the proposed D-MNN. The authors claim that this decoupling enables a better accuracy–efficiency trade-off and opens opportunities for exploring the architectural design space.

**Strengths:**

1. The core idea, progressively replacing layers while maintaining representational similarity, is intuitive yet practically effective. Although philosophically framed as a "Ship of Theseus," the method essentially defines a similarity loss to align the outputs of the replaced and original layers. This part-by-part replacement and similarity alignment strategy, while conceptually simple, is implemented systematically for the first time. Despite limited algorithmic novelty, the approach is simple, feasible, and potentially useful for future work.

2. The paper is clearly written and easy to follow.

3. It introduces a new perspective on decoupling training from deployment.

4. It shows that even untrained networks can provide transferable inductive biases, which is an interesting finding.

5. Although theoretical justification is weak, the experiments offer solid empirical evidence supporting the feasibility of the approach.

**Weaknesses:**

1. While the idea is straightforward and effective, it lacks strong novelty. Overall, the contribution lies more in the systematic implementation and empirical exploration of an intuitive idea than in a fundamentally new conceptual innovation. The four replacement schedules (progressive, sequential, independent, joint) are systematic but predictable; 'progressive' being superior is not surprising. The experiments are solid but could have explored broader domains, such as speech or multimodal settings, to better test generalization.

2. The proposed D-MNN similarity measurement is under-analyzed. CKA remains the main metric used, and D-MNN is only reported for the ResNet-18 $\rightarrow$ MLP conversion. The performance difference between D-MNN and CKA is marginal; no sensitivity to hyperparameter analysis is provided, and comparisons with other similarity measures are missing. Hence, the effectiveness of D-MNN remains insufficiently validated.

3. The paper does not discuss parameter count changes after module replacement. It only mentions shape compatibility and low-rank linear projections, but does not clarify whether parameter counts are kept constant. The absence of parameter statistics makes it difficult to judge whether performance preservation is partly due to differences in model capacity; this is an important missing piece.

4. The computational cost of NoT (training time, GPU usage, etc.) is not reported, leaving its scalability unclear.

5. While representational alignment is presented as the key mechanism, the paper does not explain why it should ensure functional equivalence across architectures. The lack of theoretical grounding weakens the conceptual depth of the work.

6. The paper does not specify how the correspondence between guide and target components is determined. The replacement mapping
$\mathcal{R}$ seems to be manually defined based on shape compatibility or pre-specified block grouping (e.g., 4 ResNet-50 blocks $\rightarrow$ 1 ResNet-18 block). However, different mapping strategies could substantially affect representational alignment and final performance. The lack of ablations or analysis on this aspect limits the understanding of the robustness and generality of the proposed method.

**Questions:**

1. The "replacement mapping" mechanism appears to be critical for the model's evolution process and might heavily influence the reported improvements. Is this understanding correct? If not, could the authors provide ablation or evidence showing the specific effect of replacement mapping on model performance?

2. See also the Weaknesses.

---

> ### Author Response · Authors · 2025-11-20
> **Response 1/3**
>
> We thank the reviewer for their thoughtful review. We are glad the reviewer found the paper “clearly written”, found the method intuitive yet practically effective, and found our findings with untrained networks interesting. We hope to address concerns below.
>
> > W1: contribution lies more in the systematic implementation and empirical exploration of an intuitive idea than in a fundamentally new conceptual innovation
>
> Thank you for the thoughtful feedback. We aim to add clarity to the novelty of our approach. The main goal of our work is to enable cross‑architecture conversion. We enable this conversion using representational similarity stage‑by‑stage, in reference to the Ship of Theseus. While other references have discussed staged procedures, none have focused on architectural transfer that is as “radical” (Reviewer ioot) as the one we are proposing. For concreteness, our experiments include conversions such as ResNet-18→MLP, ViT→Patch‑MLP, GPT‑2→RNN, and a group‑wise, non‑shape‑preserving mapping from ResNet‑50 to ResNet‑18. These represent substantial architectural shifts rather than minor variants.Furthermore, we believe our finding with untrained architectures is novel and non-obvious. We recover substantial performance starting from an untrained guide, and our method does not require labels, unlike prior work [1]. This has strong potential for new initializations for these architectures that have not been discovered before.
>
> We would also note that the fact that this is an intuitive idea does not diminish the novelty of our paper; rather, it strengthens it. To us, there is also major conceptual innovation as well, e.g. decoupling the training architecture from the inference architecture and introducing D‑MNN as an alignment metric. Regarding the replacement schedules: while it may seem predictable that progressive would be strong, the ordering and magnitude of the differences were not a foregone conclusion. Our comparisons quantify that progressive materially outperforms joint, independent, and sequential schedules across cross‑family conversions, which we view as part of the contribution. We agree that exploring speech or multimodal domains would further test generalization. The procedure itself is task‑agnostic—alignment uses unlabeled inputs, and the final fine‑tuning can target any downstream objective. We view broader domains as a natural next step.
>
> > W2:  no sensitivity to hyperparameter analysis is provided, and comparisons with other similarity measures are missing.
>
> Apologies, we should have included this. We dedicate some space to this discussion in an updated version of the paper (see Appendix Section H) and include a brief discussion here.
>
> To explore this further, we add some plots of our alignment scores that vary the temperature. Since we are finding that better alignment gives better results, our goal is to minimize our D-MNN metric.  We only report the final D-MNN loss value rather than the final training performance due to time and compute constraints. We will also perform sweeps over temperature and batch size. We show results in Appendix Section H, as well as show the plot over temperature [here](https://ibb.co/kvVfsr2)
>
> We generally don’t find significant sensitivity to temperature as we vary over orders of magnitude of temperature. We find that around 1, the temperature is less sensitive, and as we increase the temperature, the alignment gets worse. This makes sense, since higher temperatures make the distribution flatter over the top-k, which may have less signal.

---

> ### Author Response · Authors · 2025-11-20
> **Response 2/3**
>
> > W3: The paper does not discuss parameter count changes after module replacement. It only mentions shape compatibility and low-rank linear projections, but does not clarify whether parameter counts are kept constant.
>
> Thank you for raising this point. In our work, parameter counts are not kept constant. They change as needed to enable conversion across architectures. For example, when converting from a convolutional layer to a linear layer, we flatten the convolutional activations, which increases the number of parameters in the linear module. We will provide the parameter counts for each setting.
>
> | Experiment                  | Guide Network Parameters | Target Network Parameters |
> |-----------------------------|--------------------------|---------------------------|
> | ResNet-18 --> MLP           | 11.6M                    | 4.5B                      |
> | DINOv2 --> PatchMLP         | 86.5M                    | 121.9M                    |
> | GPT-2 --> RNN               | 117M                     | 124.4M                    |
> | ResNet-50 --> ResNet-18     | 25.5M                    | 11.6M                     |
> | GPT-2 Large --> GPT-2 Small | 774M                     | 117M                      |
>
> As noted above, in some cases, we increase the number of parameters in the target network. This increase is paired with a baseline trained from scratch at the same target configuration to compare against NoT. Accordingly, we believe we have controlled for improvements due solely to increased parameter counts. This is also now paired with our distillation experiment. Our goal was to demonstrate that we can transfer inductive biases across architectures.
>
> > W4: The computational cost of NoT (training time, GPU usage, etc.) is not reported, leaving its scalability unclear.
>
> We discuss this here. For each experiment, we report the maximum GPU usage and training time, as well as which GPU we used for the particular experiment.
>
> | Experiment                  | GPU Type | Replacements | Max GPU VRAM Usage (Training, GB) | Wall-Clock Training Time (Hours) |
> |-----------------------------|----------|--------------|-----------------------------------|----------------------------------|
> | ResNet-18 --> MLP           | H100     | 20           | 81.03                             | 124                              |
> | DINOv2 --> PatchMLP         | H100     | 12           | 51.64                             | 110                              |
> | GPT-2 --> RNN               | H100     | 12           | 80.95                             | 114                              |
> | ResNet-50 --> ResNet-18     | H100     | 8            | 23.54                             | 56                               |
> | GPT-2 Large --> GPT-2 Small | H200     | 12           | 114.33                            | 125                              |
>
> We note that all our models fit on H100s, other than converting GPT-2 Large to GPT-2 Small. We also note that replacement varies by the number of replacements. Faster versions of NoT can work with a less fine-grained replacement schedule, where we replace fewer components rather than replace individual layers.

---

> ### Author Response · Authors · 2025-11-20
> **Response 3/3**
>
> > W5: The lack of theoretical grounding weakens the conceptual depth of the work.
>
> This is a fair point. While we don’t have a formal proof for the key mechanism behind representational alignment in NoT, we aim to provide some intuition. We refer to prior works that establish that we are transferring inductive biases via kernel alignment. For example, [2] investigates how a network’s neural tangent kernel (NTK) aligns with a target output during training. The paper shows that NTK alignment accelerates convergence and lowers generalization error in deep linear networks. This aligned kernel condition is inserted by hand in NoT guidance. [3] is a new paper that considers task‑aware representational alignment. Their theory provides a generalization bound via kernel alignment. They show that when a “stitcher” maps representations of a source network to a target output, the excess risk of the stitched model is upper‑bounded by the CKA alignment between them. This provides a learning‑theoretic guarantee that the CKA term in NoT’s guidance reduces the set of possible hypotheses seen by an optimizer. Overfitting or underfitting becomes harder. Similarly, [4] uses Rademacher complexity tools to show that alignment of tangent‑kernel features onto a small set of task‑relevant directions compresses the effective model class. This discussion is also included in [1] as an explanation for the author’s work there. This formalizes the notion of NoT guidance as an automatic regularizer, where task directions are replaced by the guide network settings. We hope that the above covers a theoretical gap. We believe that CKA bounds the risk or complexity in terms of kernel alignment. The NTK and Rademacher analyses show that alignment shrinks the effective hypothesis space and improves conditioning. This could be sharpened by changing the alignment used in guidance—e.g., moving from aligning on kernels to aligning on singular vectors or eigenvectors instead. We hope to investigate this further in future work to provide more formal theoretical grounding.
>
> > W6/Q1: The paper does not specify how the correspondence between guide and target components is determined. The replacement mapping seems to be manually defined based on shape compatibility or pre-specified block grouping
>
> Thank you for pointing this out. In NoT, the correspondence is an explicit input to the method: the algorithm takes a replacement mapping $R$ and is otherwise agnostic to how $R$ is chosen beyond shape‑compatible connections or group-wise mappings. For clarity, we used simple, canonical choices: one‑to‑one replacements where possible (e.g., Conv→Linear, Attention→Patch‑MLP) and a stage‑aligned group mapping for ResNet‑50→ResNet‑18 (four bottleneck blocks within a stage mapped to one basic block) to respect resolution boundaries.
> While alternative mappings could be explored, several results suggest that our results are not fragile to the specific choices of $R$: 1. The progressive procedure shows insensitivity to replacement order. 2. Our diagnostics indicate that difficulty concentrates at downsampling boundaries, which our mapping preserves. 3. Across four radically different conversions, NoT consistently improves over naïve replacement. Together, these support the robustness of the approach. Given the current constraints, we could not run such ablations, but we will aim to incorporate them as well as to validate our intuition.
>
> [1] Subramaniam et. al. Training the Untrainable: Introducing Inductive Bias via Representational Alignment. NeurIPS 2025.
>
> [2] Shan and Bordelon. A Theory of Neural Tangent Kernel Alignment and Its Influence on Training, Preprint. 2022.
>
> [3] Insulla et. al. Towards a Learning Theory of Representation Alignment. Preprint, 2025.
>
> [4] Baratin, George, et. al. Implicit Regularization via Neural Feature Alignment. AISTATS, 2021.

---

> > ### Comment · Area_Chair_pm6X · 2025-11-26
> >
> > Dear reviewer fJRw,
> >
> > Could you take a look at the authors response and leave your feedback.
> >
> > AC

---

### Official Review · Reviewer_ioot · 2025-11-08

**Soundness:** 2
**Presentation:** 3
**Contribution:** 2
**Rating:** 4
**Confidence:** 3

**Summary:**

This paper introduces Network of Theseus (NoT), a method for converting neural network architectures by progressively replacing components while maintaining representational alignment. NoT enables dramatic architectural transformations such as converting CNNs to MLPs, ViTs to token-wise MLPs, and Transformers to RNNs via optimising intermediate layer activations using similarity metrics like Centred Kernel Alignment (CKA). The key innovation lies in staged progressive replacement that minimises error accumulation. Empirical results are reported for several substantial cross-architecture transformations on datasets like ImageNet and Wikitext-103, demonstrate that NoT achieves substantial performance improvements over naive replacement baselines, and works even with untrained guide networks.

**Strengths:**

1. Unlike prior works, NoT is not limited to structurally similar architecture or reliance on identical computational patterns, e.g., attention to linear attention, but can handle radical family shifts, e.g., ResNet to MLP, GPT-2 to RNN, etc.

2. The progressive replacement strategy is quite elegant and well-justified. The "Ship of Theseus" metaphor effectively communicates the core idea of this work. The proposed method addresses a limitation in the current neural architecture research, the tight coupling between training and deployment architectures.

3. The ablation studies are rigorous, authors contrast progressive, joint, independent, and sequential replacement, robustly arguing for the effectiveness and stability of progressive alignment.

**Weaknesses:**

1. The main baselines are naive replacement and training from scratch, but there are no strong comparisons to SoTA methods, such as in progressive distillation, model stitching, and neural architecture search that also attempt cross-architecture or sub-graph-level transfer. For example, some relevant works have been briefly discussed and cited in the paper,  but there is no direct comparison in the main results.

2. The tuning of hyper-parameters for the D-MNN metric, such as temperature choice, batch size effects, and differentiable kNN mechanisms is only briefly discussed. Implementation guidance and impact on downstream results are underexplored, especially for D-MNN sometimes yields comparable but lower results than CKA.

3. Although authors claim that the NoT is a general method, the progressive replacement requires multiple stages of optimisation, each involving careful learning rate tuning, batch normalisation recalibration, and multiple random seed runs, this could undermines the practical applicability.

**Questions:**

1. Can authors provide some extra information such as wall-clock time or GPU hours, for the computational cost of NoT, compared to training from scratch?

2. What determines when to stop optimising at each stage? Is there a principled way to detect convergence of representational alignment?

3. Can authors discuss a bit on the failure cases? under what conditions does NoT fail?

4. Regarding the batch normalisation, can author quantify the impact of the batch normalisation handling strategy? What happens if you use Layer Normalisation or Group Normalisation instead?

---

> ### Author Response · Authors · 2025-11-20
> **Response 1/3**
>
> We thank the reviewer for their thoughtful review. We are glad the reviewer appreciated the generality of our approach (“NoT is not limited to structurally similar architecture”), found the method “elegant” and “well-justified”, and found the ablation studies “rigorous”. We hope to address concerns below.
>
> > W1: The main baselines are naive replacement and training from scratch, but there are no strong comparisons to SoTA methods, such as in progressive distillation
>
> Thank you. We believe progressive knowledge distillation falls under the scope of approaches in our paper since the method involves staged replacement with an alignment metric, which is aligning output logits. To the best of our knowledge, distillation has not been applied across large architectural shifts. To demonstrate our novelty further, we include a comparison with progressive distillation. In this comparison, we rerun our experiments with a progressive notion of distillation, where at each stage, we align the layers while running a distillation loss function. In our progressive distillation setting, we replace a patch of our original network with our target architecture and update only the replaced component using knowledge distillation rather than using representational similarity. The other components of the network are frozen until we apply standard training end-to-end. We show the results below.
>
> |         | Guide | NoT Target | Distillation Target | Original |
> |-------------------------------------|-------|------------|---------------------|----------|
> | ResNet-18$\rightarrow$MLP           | 69.16 | 62.55      | 59.13    | 33.36    |
> | Untrained ResNet-18$\rightarrow$MLP | 0.10  | 60.28      | 31.11    | 33.36    |
> | GPT-2$\rightarrow$RNN ($\downarrow$)               | 37.50 | 50.58      | 75.49   | 121.19   |
> | Untrained GPT-2$\rightarrow$RNN ($\downarrow$)    |   51914.5    | 58.26   | 135.55     | 121.19   |
>
> > W2: The tuning of hyper-parameters for the D-MNN metric
>
> This is fair. We dedicate some space to this discussion in an updated version of the paper (see Appendix Section H) and include a brief discussion here.
>
> To explore this further, we add some plots of our alignment scores that vary the temperature. Since we are finding that better alignment gives better results, our goal is to minimize our D-MNN metric. We show results at this [link](https://ibb.co/kvVfsr2 ) and in Appendix Section H. In general, we find that close temperatures don’t have too much variance in results, i.e., between 0.5-2, but higher temperatures show larger variation in results with lower alignment. This makes sense, since higher temperatures make the distribution flatter over the top-k, which may have less signal.
>
> We generally find that, like CKA, D-MNN is sensitive to learning rates and batch sizes. Bigger batch sizes generally lead to better similarity of representations in our current implementation. We will dedicate additional experiments demonstrating sensitivity to learning rates and batch sizes in Appendix Section H, but show temperature as our first analysis.

---

> > ### Author Response · Authors · 2025-11-20
> > **Response 2/3**
> >
> > > W3: the progressive replacement requires multiple stages of optimisation, each involving careful learning rate tuning, batch normalisation recalibration, and multiple random seed runs
> >
> > We hope to address concerns about the expensive nature of the progressive replacement here. BN recalibration in NoT is a single, post‑alignment step performed before standard task training on architectures that use BN. In practice, this is lightweight. Moreover, many modern architectures (e.g., Transformers) rely on layer normalization, so the impact of BN handling is limited in those settings. In future experiments, we will explore replacing the convolution+BN stack with a linear layer plus layer normalization, which could remove BN tuning entirely. We will quantify the effect on accuracy as well in response to your specific question.
> >
> > We agree that progressive replacement benefits from sensible stage‑wise learning rates. To reduce manual tuning, we are moving to a preset learning rate with a [reduce‑on‑plateau](https://docs.pytorch.org/docs/stable/generated/torch.optim.lr_scheduler.ReduceLROnPlateau.html) scheduler, plus EMA. Our stopping rule is simple: proceed when the EMA of the alignment loss crosses a target (e.g., CKA above 0.99 for the replaced layers) or after a cap of 100 epochs. This lets the scheduler adapt automatically, a common practice in training pipelines. We will include results from this automated recipe in the revision.  We show results from this new algorithm here and will incorporate this into the other experiments as well, but this will take a bit longer.
> >
> > |   | Guide | NoT (EMA) | NoT (Original) |
> > |---|-------|-----------|----------------|
> > | Wikitext-103 Perplexity ($\downarrow$)  | 37.50 | 48.56     | 50.58          |
> >
> >
> > We observe that the removal of manual tuning and sensitivity to the dynamics of the loss curve slightly improves results.
> >
> > While progressive replacement gives the strongest results overall, our joint schedule (all replacements optimized together) still provides a large improvement over naïve training and uses a single learning rate. We can also interpolate between the two—replacing groups of layers per stage—which reduces the number of stages while retaining much of the benefit. This grouped strategy is what we effectively do in the ResNet‑50 → ResNet‑18 conversion, and it improves over baseline while requiring fewer stages. We will add these details in the paper’s discussion of schedules.

---

> > > ### Author Response · Authors · 2025-11-20
> > > **Response 3/3**
> > >
> > > > Q1: Can authors provide some extra information such as wall-clock time or GPU hours
> > >
> > > We discuss computational cost. For each experiment, we report the maximum GPU usage and training time, as well as which GPU we used for the particular experiment.
> > >
> > > | Experiment                  | GPU Type | Replacements | Max GPU VRAM Usage (Training, GB) | Wall-Clock Training Time (Hours) |
> > > |-----------------------------|----------|--------------|-----------------------------------|----------------------------------|
> > > | ResNet-18 --> MLP           | H100     | 20           | 81.03                             | 124                              |
> > > | DINOv2 --> PatchMLP         | H100     | 12           | 51.64                             | 110                              |
> > > | GPT-2 --> RNN               | H100     | 12           | 80.95                             | 114                              |
> > > | ResNet-50 --> ResNet-18     | H100     | 8            | 23.54                             | 56                               |
> > > | GPT-2 Large --> GPT-2 Small | H200     | 12           | 114.33                            | 125                              |
> > >
> > > We note that all our models fit on H100s, other than converting GPT-2 Large to GPT-2 Small. We also note that replacement varies by the number of replacements. Faster versions of NoT can work with a less fine-grained replacement schedule, where we replace fewer components rather than replace individual layers.
> > >
> > > > Q2: What determines when to stop optimising at each stage? Is there a principled way to detect convergence of representational alignment?
> > >
> > >
> > > This is a good question. In theory, we can’t predict convergence of representational alignment. In practice, we find that a CKA alignment of 0.99 is a useful heuristic for stopping once we have optimised long enough. While this threshold is not reached at some stages (0.99 CKA is quite high), we believe it approximates our results well and alleviates associated difficulties with optimizing similarity.
> > >
> > > > Q3: Can authors discuss a bit on the failure cases? under what conditions does NoT fail?
> > >
> > > Yes!  In particular, we highlight cases where untrained architectures generally fail to recover performance of the target architecture. One such case we have identified is converting ResNet‑50 to a deep ConvNet with no residual connections. Residual connections [1] were introduced as part of training stability to address issues with gradient credit assignment in very deep networks. Demonstrating that residual connections can be removed would have interesting implications for understanding the function of residual connections, i.e., do they have an effect on the representation space of ResNet‑50 or act as a stabilizer during optimization?
> > >
> > >
> > > We show results below:
> > >
> > > |     | Guide | NoT Target | Original Performance |
> > > |---------------------------------------|-------|------------|----------------------|
> > > | RN-50$\rightarrow$Deep Conv           | 76.13 | 72.11      | 67.19     |
> > > | Untrained RN-50$\rightarrow$Deep Conv | 76.13 | 65.12      | 67.19     |
> > >
> > >
> > > We find that we don’t improve using an untrained ResNet‑50 guide network as our original network. We believe this demonstrates a useful failure case where we cannot transfer inductive biases to an architecture that shares said inductive biases. This is also reflected in another experiment where we convert an MLP → MLP and see no improvement. When we convert from our uninitialized MLP to the same MLP architecture, we do not see an improvement in downstream MLP performance. This indicates that representational similarity does not act as a basic regularizer on the target architecture.
> > >
> > > One additional aspect we are actively investigating is whether architectures generalize after using NoT. For example, if we transfer from ResNet‑18 to an MLP, does the resultant MLP generalize to new tasks? Unfortunately, the time constraint of this rebuttal period prevents us from carrying out the relevant experiment, but we aim to investigate this as soon as possible.
> > >
> > > > Q4: Regarding the batch normalisation, can author quantify the impact of the batch normalisation handling strategy?
> > >
> > > We would like to emphasize that batch normalization is an implementation detail rather than an expensive technique. The normalization is only done for 200 steps to renormalize. This helps reduce activation shift during standard training and is important for trained models. In total, this takes less than one minute. We find that we drop by 10% in the trained network case when we do not renormalize because the network underfits. This is the opposite problem from the original MLP, which would traditionally overfit during training on ImageNet. BN recalibration also helps accelerate convergence of standard training.

---

> > > > ### Comment · Area_Chair_pm6X · 2025-11-26
> > > >
> > > > Dear reviewer ioot,
> > > >
> > > > Could you take a look at the authors response and leave your feedback.
> > > >
> > > > AC

---

> > > > ### Comment · Reviewer_ioot · 2025-11-27
> > > >
> > > > First of all, I'd like to thank the authors for their extra work and clarifications, which partially addressed my concerns. However, 1) the empirical comparisons are still a bit narrow; 2) although the authors mentioned an automated schedule, I couldn't find this improvement in the revised paper. Right now, the method still looks extremely expensive and difficult to apply; 3) the authors reported hardware and  computational costs, which is good, but it also reinforces my concern regarding the practicability. Whether the benefits offered by NoT surpassed its computational overhead remains a question.
> > > >
> > > > Due to above reasons, I lean to maintain my rating at this stage.

---

> ### Author Response · Authors · 2025-11-29
>
> We would like to thank the reviewer for their engagement with our paper. We hope to address further concerns here.
>
> > Empirical comparisons are still a bit narrow
>
> Thank you. We have added ConvNeXt [1] as a new architecture to convert from as part of NoT. In this setting, we convert every 1x1 convolutional layer to a linear layer to create a Patch MLP, like with DINOv2 with no token mixing. We will incorporate this as part of the paper and have added this to Appendix Section J.  We hope that we have demonstrated NoT on a wide array of architectures from convolutional networks to vision transformers to language models. See ConvNeXt results below.
>
>
>
> | Experiment             | Original | NoT   | Baseline |
> |------------------------|----------|-------|----------|
> | ConvNeXt --> Patch MLP | 82.22    | 64.14 | 58.43    |
>
> > Automated schedule
>
> We apologize for the oversight. We have added the automated schedule to the paper in Appendix Section K. We have a description of the methodology as well as our current results associated. We hope that our automatic scheduling shows our method can make our method simpler to apply. More than this, we emphasize that our method is flexible with respect to the number of replacements in the schedule. One can change the number of replacements in the schedule to reduce run time.
>
> Overall, we would like to emphasize our effort to address comments from the reviewer. We have added numerous experiments like new settings for NoT with GPT-2 Large and ConvNeXt, reported hardware usage and made modifications to make NoT more automatic to run. Requested baselines like progressive distillation were also added. We hope that the effort of responses be taken into consideration now that reviewer responses have been muted.
>
> [1] Liu et. al. A convnet for the 2020s. CVPR 2022.

---

### Author Response · Authors · 2025-11-20
**General Response**

We thank the reviewers for their thoughtful reviews. We are glad reviewers found the method in our paper interesting (5FPe), intuitive yet practical, introducing radical architectural shifts (fjRw, ioot), found the paper clearly written (zw5v, fjRw), and appreciated our thorough experiments (5FPe, zw5v). Along with addressing several comments and concerns, we would also like the chance to highlight several additional experiments and changes we have introduced in the manuscript as well. All changes are highlighted in blue text. We summarize our changes as well below.

New experiments:

* D-MNN on all settings: We reran all settings with D-MNN as a similarity metric to establish whether D-MNN is a generally performant metric. We found that D-MNN significantly improved over baseline performance in all settings. We believe this strengthens our claims for D-MNN as a new similarity metric that can be optimized to transfer properties across neural networks.
* GPT-2 Large → GPT-2 Small: We introduced new experiments where we converted GPT-2 Large (774M) to GPT-2 Small (117M) using the same approach as applied for ResNet-50 → ResNet-18. We find similar improvements and incorporate this into our Tables 1-3 in the paper. As reported with ResNet-50 → ResNet-18, we also find improvements from using an untrained architecture as our guide network. We would like to emphasize the significance of this finding: to our knowledge, prior work has not established exploiting untrained, randomly initialized architectures in this manner. For this conversion, we used unbiased CKA [1], which we discuss in Appendix Section C.1.1 and Appendix Section D in the paper. In our experiments, we found that biased CKA could not distinguish between representations of GPT-2 and GPT-2 Large very well, and found that the HSIC estimator for CKA was biased. This led us to choose the unbiased estimator, which we discuss in the paper.

Reviewer Comments:

* D-MNN hyperparameter sensitivity (ioot, fJRw): Along with our previous expansion of D-MNN to all settings, we include a hyperparameter sweep of D-MNN to establish sensitivity to parameters such as learning rate or temperature. We find that our formulation is not very sensitive to temperature when we measure alignment until much larger temperature values, which flatten the distributions. This is reported in Appendix Section H, and the plot is included [here](https://ibb.co/kvVfsr2).
* Distillation Comparison (ioot, 5FPe, zw5V): We introduce a comparison with progressive distillation, where we convert every layer progressively and run knowledge distillation  to train the converted layers. We apply this for both ResNet-18 → MLP and GPT-2 → RNN as a comparison across both vision and language tasks. We find improved performance from using a trained guide network but reduced performance from using an untrained guide architecture. This is reported in Appendix Section I.  We believe progressive knowledge distillation falls under the scope of approaches in our paper since the method involves staged  replacement with an alignment metric, which is aligning output logits. To the best of our knowledge, distillation has not been applied across radical architectural shifts.

|         | Guide | NoT Target | Distillation Target | Original |
|-------------------------------------|-------|------------|---------------------|----------|
| ResNet-18$\rightarrow$MLP           | 69.16 | 62.55      | 59.13    | 33.36    |
| Untrained ResNet-18$\rightarrow$MLP | 0.10  | 60.28      | 31.11    | 33.36    |
| GPT-2$\rightarrow$RNN ($\downarrow$)               | 37.50 | 50.58      | 75.49   | 121.19   |
| Untrained GPT-2$\rightarrow$RNN ($\downarrow$)    |   51914.5    | 58.26   | 135.55     | 121.19   |

* Novelty (zw5V, fjRw): We address concerns of novelty with regard to the findings in our paper. While previous works have transferred across networks, our shifts are larger than previously proposed. We highlight our findings with untrained architectures, which are not replicated via knowledge distillation. We cover some papers in an updated related work section.

[1] Song et. al. Supervised feature selection via dependence estimation. ICML 2007

---

### Author Response · Authors · 2025-12-01
**Message to AC**

Dear AC,

We would like to thank you for your hard work. We have received valuable feedback from reviewers in this ICLR cycle. We want to take this opportunity to briefly summarize our additions to the paper and our responses to the reviewers’ concerns. All changes in the paper are shown in blue font.

To summarize our paper, we introduce the Network of Theseus (NoT), a method for progressively converting one network architecture into another part-by-part using representational similarity. We apply NoT across a range of architectures, finding that we preserve a significant percentage of performance even when the starting architecture is untrained.

> Newer architectures (5FPe)

Some reviewers have pointed out that the architectural conversions we apply in the paper are older. To address this, we added newer language model conversions like converting GPT-2 Large to GPT-2 Small. We also recently added ConvNeXt, where we convert every convolutional layer to a patch-wise MLP, similar to DINOv2. We find that performance is well preserved using NoT. These updates are now integrated into Sections 4 and 5 as well as Appendix Section J.

> D-MNN Exploration (ioot, fJRw)

Some reviewers asked for further analysis of our new metric, D-MNN. We have conducted the requested analysis by varying hyperparameters such as temperature and reported the results in Appendix Section H. We find that D-MNN is not very sensitive to temperature.

> Progressive Distillation baseline (ioot, 5FPe, zm5V)

Several reviewers requested a comparison with knowledge distillation. We have applied knowledge distillation progressively, per stage, and found that our method works significantly better, especially with untrained networks. See Appendix Section I.

> How expensive is NoT (ioot, fJRw)

As requested by the reviewers, we report all GPU VRAM usage and wall clock time for applying NoT. We also cover an automated schedule to make NoT significantly cheaper, where we tune every stage by reducing the learning rate when the similarity loss plateaus and introducing a new stopping condition. We find that this improves results when converting from GPT-2 to an RNN. See Appendix Section K.

> Novelty (5FPe, zw5V)

In the latest version of the paper, we emphasize the novelty and contributions of our approach. Please review the sections highlighted in blue.

Thank you again for reviewing our paper!

---

### Meta-Review · Area_Chair_g7nS · 2025-12-24

**Summary:**

In this work, the authors challenge the inductive bias assumption, i.e., the architecture used in training must be the same as the one at inference. The authors propose Network of Theseus (NoT), which starts at a guide network and progressively replace part-by-part to a target architecture using representational similarity metrics, involving centered kernel alignment (CKA) and differential mutual nearest neighbors (D-MNN).

The Reviewers agree that the core idea of progressively replacing part-by-part and maintaining representation similarity, is effective for the considered problem. However, the Reviewers raised concerns on experimental setup involving hyperparameters, baselines; its novelty; complexity; relation between representation similarity with functional equivalence. In the rebuttal, the authors provide additional empirical results, more intuition for the proposed approach; improve the presentation.

Overall, we think the submission falls short to the bar. The authors may improve the submission following the Reviewers' suggestions. Although the proposed ideas are interesting and potential, more analysis and major revision are required.

**Reviewer Concerns:**

Concerns from the Reviewers are as follows:

+ Reviewer ioot: weak experimental setup; lacking guidance on hyperparameters influences; difficulty of multi-stage optimization; extremely expensive computation.

+ Reviewer fJRw: weak theoretical justification; novelty; weak evaluation for similarity metric; hyperparameters sensitivity; analysis on model capacity/parameters; computational complexity; lack of theoretical grounding for relation between representation similarity and functional equivalence; replacement mapping.

+ Reviewer zw5V: presentation; related work; weak experimental setup; lack of hyperparameters ablation

+ Reviewer 5FPe: more rigorous emphasis on result impact; novelty; weak experimental setup.

**Reviewer Scores:**

The authors provide additional empirical results; further intuition for the design; improve the presentation in the updated version. We think the authors may partially address some concerns of the Reviewers. Several concerns involving complexity, experimental setup and analysis may not be convincing enough yet. Consequently, although the core idea is interesting, a major revision and another round of review are necessary.

---

### Decision · Program_Chairs · 2026-01-26

Reject